# Neuromodulatory connectivity defines the structure of a behavioral neural network

**Feici Diao[1], Amicia D Elliott[2], Fengqiu Diao[1], Sarav Shah[1], Benjamin H White[1]\***

[1]Laboratory of Molecular Biology, National Institute of Mental Health, National Institutes of Health, Bethesda, United States; [2]National Institute of General Medical Sciences, National Institutes of Health, Bethesda, United States

**Abstract** Neural networks are typically defined by their synaptic connectivity, yet synaptic wiring diagrams often provide limited insight into network function. This is due partly to the importance of non-synaptic communication by neuromodulators, which can dynamically reconfigure circuit activity to alter its output. Here, we systematically map the patterns of neuromodulatory connectivity in a network that governs a developmentally critical behavioral sequence in *Drosophila*. This sequence, which mediates pupal ecdysis, is governed by the serial release of several key factors, which act both somatically as hormones and within the brain as neuromodulators. By identifying and characterizing the functions of the neuronal targets of these factors, we find that they define hierarchically organized layers of the network controlling the pupal ecdysis sequence: a modular input layer, an intermediate central pattern generating layer, and a motor output layer. Mapping neuromodulatory connections in this system thus defines the functional architecture of the network.

DOI: https://doi.org/10.7554/eLife.29797.001

**\*For correspondence:**
benjaminwhite@mail.nih.gov

**Competing interests:** The authors declare that no competing interests exist.

## Introduction

Neuromodulators constitute a major channel of communication in the nervous system and act at virtually all levels of sensorimotor processing to tune the intrinsic and synaptic properties of neurons (*Marder, 2012*; *Nadim and Bucher, 2014*; *van den Pol, 2012*). How these properties are tuned can profoundly influence the function of not only the individual neurons, but also the circuits in which they participate. Dynamic and coordinated regulation of multiple brain circuits is required for behavior, and an attractive idea with deep historical roots is that neuromodulators, which typically act over timescales and distances that are long compared with synaptic neurotransmission, may serve to coordinate activity in broadly distributed circuits to bias the performance of behaviors appropriate for a given set of circumstances (*Bargmann, 2012*; *Bicker and Menzel, 1989*; *Harris-Warrick and Marder, 1991*). Although the generality and scope of this viewpoint remain unclear, it implies that identifying sites of neuromodulator action may represent a productive strategy for mapping critical circuits involved in generating a behavior of interest.

The strategy of mapping sites of neuromodulator action emerged from early observations on the activating effects of various brain-derived hormones and biogenic amines. When introduced into the nervous system, these compounds were found to induce complex motor programs (*Harris-Warrick, 1988*), such as emergence of moths from their cocoons (*Truman and Riddiford, 1970*), egg-laying in sea hares (*Kupfermann, 1967*), and postural changes in lobsters (*Livingstone et al., 1980*). Attempts by *Sombati and Hoyle (1984)* to map the sites of action of the insect locomotor activator, octopamine (OA), in the metathoracic ganglion of locusts led to Hoyle's 'orchestration hypothesis,' according to which flight and other motor programs are encoded in the activities of

**eLife digest** Why do animals behave the way they do? Behavior occurs in response to signals from the environment, such as those indicating food or danger, or signals from the body, such as those indicating hunger or thirst. The nervous system detects these signals and triggers an appropriate response, such as seeking food or fleeing a threat. But because much of the nervous system takes part in generating these responses, it can make it difficult to understand how even simple behaviors come about.

One behavior that has been studied extensively is molting in insects. Molting enables insects to grow and develop, and involves casting off the outer skeleton of the previous developmental stage. To do this, the insect performs a series of repetitive movements, known as an ecdysis sequence. In the fruit fly, the pupal ecdysis sequence consists of three distinct patterns rhythmic abdominal movement. A hormone called ecdysis triggering hormone, or ETH for short, initiates this sequence by triggering the release of two further hormones, Bursicon and CCAP. All three hormones act on the nervous system to coordinate molting behavior, but exactly how they do so is unclear.

Diao et al. have now used genetic tools called Trojan exons to identify the neurons of fruit flies on which these hormones act. Trojan exons are short sequences of DNA that can be inserted into non-coding regions of a target gene to mark or manipulate the cells that express it. When a cell uses its copy of the target gene to make a protein, it also makes the product encoded by the Trojan exon. Using this technique, Diao et al. identified three sets of neurons that produce receptor proteins that recognize the molting hormones. Neurons with ETH receptors start the molting process by activating neurons that make Bursicon and CCAP. Neurons with Bursicon receptors then generate motor rhythms within the nervous system. Finally, neurons with CCAP receptors respond to these rhythms and produce the abdominal movements of the ecdysis sequence.

Many other animal behaviors depend on substances like ETH, Bursicon and CCAP, which act within the brain to change the activity of neurons and circuits. The work of Diao et al. suggests that identifying the sites at which such substances act can help reveal the circuits that govern complex behaviors.

DOI: https://doi.org/10.7554/eLife.29797.002

subpopulations of OA neurons (*Hoyle, 1985*). Although the explicit statement of this influential hypothesis remains to be proven, a generalized role for OA in coordinating insect flight was supported by subsequent discoveries that OA neurons modulate muscle metabolism and visual motion processing during flight (*Libersat and Pflueger, 2004*; *Suver et al., 2012*).

Mechanistic insight into how neuromodulators regulate and coordinate circuit function came from the intensive functional and anatomical investigation of small circuits (*Bargmann and Marder, 2013*; *Selverston, 2010*). This work included fine-mapping sites of neuromodulator action by painstaking physiological characterization of single neuron responses in, among other systems, the crustacean stomatogastric ganglion (*Flamm and Harris-Warrick, 1986*; *Hooper and Marder, 1987*; *Swensen and Marder, 2001*). The STG houses two principal central pattern generators (CPGs) that drive digestive rhythms and their activity patterns, the pyloric and gastric mill rhythms, are both dependent upon, and can be variously reconfigured by, the actions of neuromodulators (*Marder and Bucher, 2007*). Two of these, the neuropeptides proctolin and *C. borealis* Tachykinin-Related Peptide Ia (CabTRP), offer a simple example of how neuromodulators acting at different sites can coordinate changes in two (overlapping) circuits (*Nusbaum, 2002*; *Nusbaum et al., 2001*). Both proctolin and CabTRP are released from a neuromodulatory projection neuron (MCN1) into the STG (*Blitz et al., 1999*), and although both peptides activate the same inward current (*Swensen and Marder, 2000*), their effects on the pyloric and gastric mill rhythms differs because they target distinct cells within the respective CPGs (*Blitz et al., 1999*; *Wood et al., 2000*). Proctolin principally excites the pyloric circuit and can activate it from quiescence, while CabTRP is required for the gastric mill rhythm and acts on a key neuron in its CPG in addition to neurons of the pyloric circuit. Activation of MCN1 by mechanosensory inputs from the stomach, induces a gastric mill rhythm via the action of CabTRP and alters the pyloric rhythm in response to the actions of both

peptides (*Beenhakker and Nusbaum, 2004*; *Blitz et al., 2004*). Sensory information, conveyed by two neuromodulators, thus produces coordinated changes in two functionally related circuits.

The significance and adaptive value of many neuromodulatory effects characterized in the STG remains unknown, and, in general, the sheer abundance of circuit neuromodulation revealed by studies of this and other small systems challenges the simple idea of 'chemical coding' of behavior by neuromodulators. This complexity is also underscored by analyses of neuromodulator receptor-distributions, first undertaken by ligand autoradiography in the 1980s. On the one hand, these studies supplied strong evidence that neuromodulators could act at many sites and over long distances, but they also highlighted the difficulty of establishing which sites were relevant for performance of specific behaviors without knowledge of where and under what circumstances each neuromodulator was released (*Herkenham, 1987*). For neuromodulators already implicated in specific behaviors, however, the receptor distributions sometimes spectacularly confirmed the idea that neuromodulators target ethologically significant circuits (*Insel and Young, 2000*). For example, cross-species differences in striatal expression of vasopressin receptors in two closely related vole species were shown to correlate with, and in fact cause, monogamous and polygamous predispositions in mating (*Hammock and Young, 2005*; *Young et al., 1997*). Based on these and other examples, variations in neuromodulator receptor expression during speciation have been proposed to be a major driver of behavioral evolution (*Katz and Lillvis, 2014*).

The recent development of genetic techniques for targeting and functionally manipulating neurons in genetic model animals has facilitated the functional characterization of neuronal populations on which neuromodulators act (*Spangler and Bruchas, 2017*). This work again provides examples of neuromodulators that coordinate activity in broadly distributed circuits. The evidence is particularly compelling for conserved neuromodulators, such oxytocin (*Mitre et al., 2016*; *Stoop, 2012*; *2014*), which in mice regulates distinct circuits that promote social behaviors, including conspecific recognition (*Ferguson et al., 2001*), pup retrieval (*Marlin et al., 2015*), and social learning (*Choe et al., 2015*). Oxytocin's homologs likewise act on circuits that facilitate behaviors related to affiliation and reproduction in species as diverse as worms (*Garrison et al., 2012*), leeches (*Wagenaar et al., 2010*), fish (*Reddon et al., 2015*), and birds (*Kelly and Goodson, 2014a*). Similarly, members of the Neuropeptide Y (NPY) signaling pathway have been shown to act on circuits that promote feeding in multiple species (*Taghert and Nitabach, 2012*).

Like many neuromodulatory signaling systems, however, oxytocin, NPY, and their receptors are widely distributed in nervous systems and are likely to function in multiple contexts (*Chronwall and Zukowska, 2004*; *Kelly and Goodson, 2014b*). This added complexity in neuromodulator action, together with the observation that neuromodulators rarely, if ever, act in isolation, has made it difficult to simply generalize the conclusion that neuromodulators organize activity in broadly distributed circuits to produce adaptive changes in behavioral expression. Given the evidence that favors such an organizational role, however, it remains a potentially useful strategy to map sites of neuromodulator action to identify key circuits involved in the generation of behaviors of interest. Particularly suitable behaviors for this approach are those for which both the neuromodulators important for behavioral performance and the circumstances under which they are released are known. The hormonally governed behaviors that underlie insect molting meet these criteria (*Truman, 2005*; *Zitnan and Adams, 2012*).

Molting is accomplished by the serial execution of several motor programs in what is called an ecdysis sequence (*White and Ewer, 2014*). Ecdysis sequences are initiated by the peripheral release of Ecdysis Triggering Hormone (ETH), which facilitates the secretion of multiple other peptide hormones, including Eclosion Hormone (EH), Crustacean Cardioactive Peptide (CCAP), and Bursicon. These factors function as neuromodulators within the nervous system to orchestrate the progression of the ecdysis sequence, and their action has been extensively studied in *Drosophila* at the pupal stage (*Diao et al., 2016*; *Kim et al., 2015*; *Kim et al., 2006*; *Mena et al., 2016*). The pupal ecdysis sequence of the fly consists of three distinct behavioral phases, the second of which is governed by CCAP and Bursicon released from neurons that are targets of ETH (*Diao et al., 2015*; *Kim et al., 2015*; *Kim et al., 2006*; *Lahr et al., 2012*). Here, we use the recently developed Trojan exon method (*Diao et al., 2015*), which permits the high-fidelity targeting of neurons that express specific neuromodulator receptors, to investigate the downstream effectors of ETH, CCAP, and Bursicon and show that the sites of action of these factors expose the structure and operational logic of the fly pupal ecdysis circuit.

# Results

## Neurons expressing ETHRA and CCAP regulate all pupal ecdysis phases

Each of the three phases of the *Drosophila* pupal ecdysis sequence is characterized by a dominant abdominal motor rhythm (*Video 1*, *Kim et al., 2006*). The entire sequence can be induced by injection of ETH1 (one of two ETH peptides encoded by the *ETH* gene in *Drosophila*), and all three phases have been proposed to be under the control of distinct peptidergic neurons (*Kim et al., 2006*). This is clearly true of the second behavioral phase (Phase II, sometimes referred to as 'ecdysis'), which is specifically dependent on CCAP and the heterodimeric hormone Bursicon (*Kim et al., 2006*; *Lahr et al., 2012*).

We have previously shown that a subset of CCAP-expressing neurons (ETHRA/CCAP neurons) also expresses the A-isoform of the ETH receptor (ETHRA) and is required for head eversion, a signature event of pupal ecdysis that occurs during Phase II (*Diao et al., 2016*). The ETHRA/CCAP neurons also include the subset expressing Bursicon (*Figure 1A*), and we began our investigation by confirming that chronic suppression of these neurons using the inward rectifying channel Kir2.1 blocks execution of Phase II. Consistent with previous observations, this manipulation also inhibits execution of the third motor program (Phase III, or 'post-ecdysis') and extends the duration of the first (Phase I or 'pre-ecdysis,'; *Figure 1B*), suggesting that Phase I may be terminated by the onset of Phase II (*Kim et al., 2015*; *Mena et al., 2016*). To determine whether this is the case, we used the temperature-sensitive dTrpA1 channel to activate the ETHRA/CCAP neurons immediately after the onset of abdominal lifting, which initiates Phase I and is accompanied by rolling waves of anteriorly directed contractions of the lateral body wall that alternate from one side of the animal to the other. We observed rapid termination of Phase I and initiation of Phase II, which was then followed by execution of Phase III (*Figure 1C*). This was the case whether dTrpA1-mediated activation was sustained throughout the observation period or was transient (i.e. 1 min; *Video 2*).

The rapid termination of Phase I upon activation of ETHRA/CCAP neurons, together with the extended duration of this phase when these same neurons are suppressed, demonstrates that ETHRA/CCAP neuron activity is necessary and sufficient for normal Phase I termination in addition to Phase II initiation. Furthermore, because Phase III behaviors follow, or fail with, those of Phase II when ETHRA/CCAP neurons are activated or suppressed, respectively, we conclude that these neurons are also important determinants of Phase III. To establish how the ETHRA/CCAP neurons regulate the three phases of the pupal ecdysis sequence, we sought to identify and characterize their downstream signaling partners.

## CCAP and Bursicon target distinct groups of neurons that are essential for pupal ecdysis

Genetic data demonstrate that CCAP and Bursicon jointly mediate signaling by the ETHRA/CCAP neurons: Most animals bearing null mutations in both the *CCAP* gene and the gene encoding Pburs—one of the two subunits of Bursicon—execute Phase I, but not Phase II behaviors, and 70–90% do not evert their heads during the vigorous side-to-side swinging that characterizes Phase II

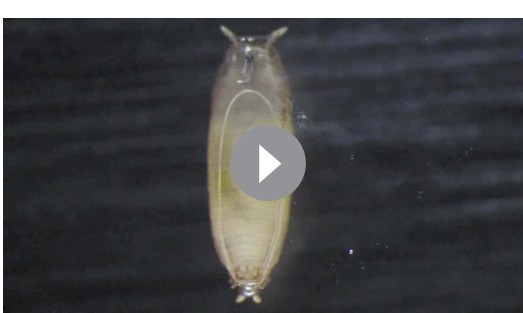

**Video 1.** The pupal ecdysis sequence and its constituent motor programs. Video speed: 20X.
DOI: https://doi.org/10.7554/eLife.29797.003

(*Lahr et al., 2012*). The severe head eversion deficits seen in *CCAP/Pburs* double mutants must result from synergistic actions of the two hormones because animals with null mutations only in *CCAP* display relatively normal pupal ecdysis behavior, and over half of animals with null mutations in *Pburs* are able to complete pupal ecdysis, although most show delays in head eversion.

To characterize the downstream neurons that mediate the effects of CCAP and Bursicon, we used the Trojan exon method to generate transgenic fly lines that express Gal4 or Split Gal4 components specifically in cells that express either the Bursicon receptor (encoded by the

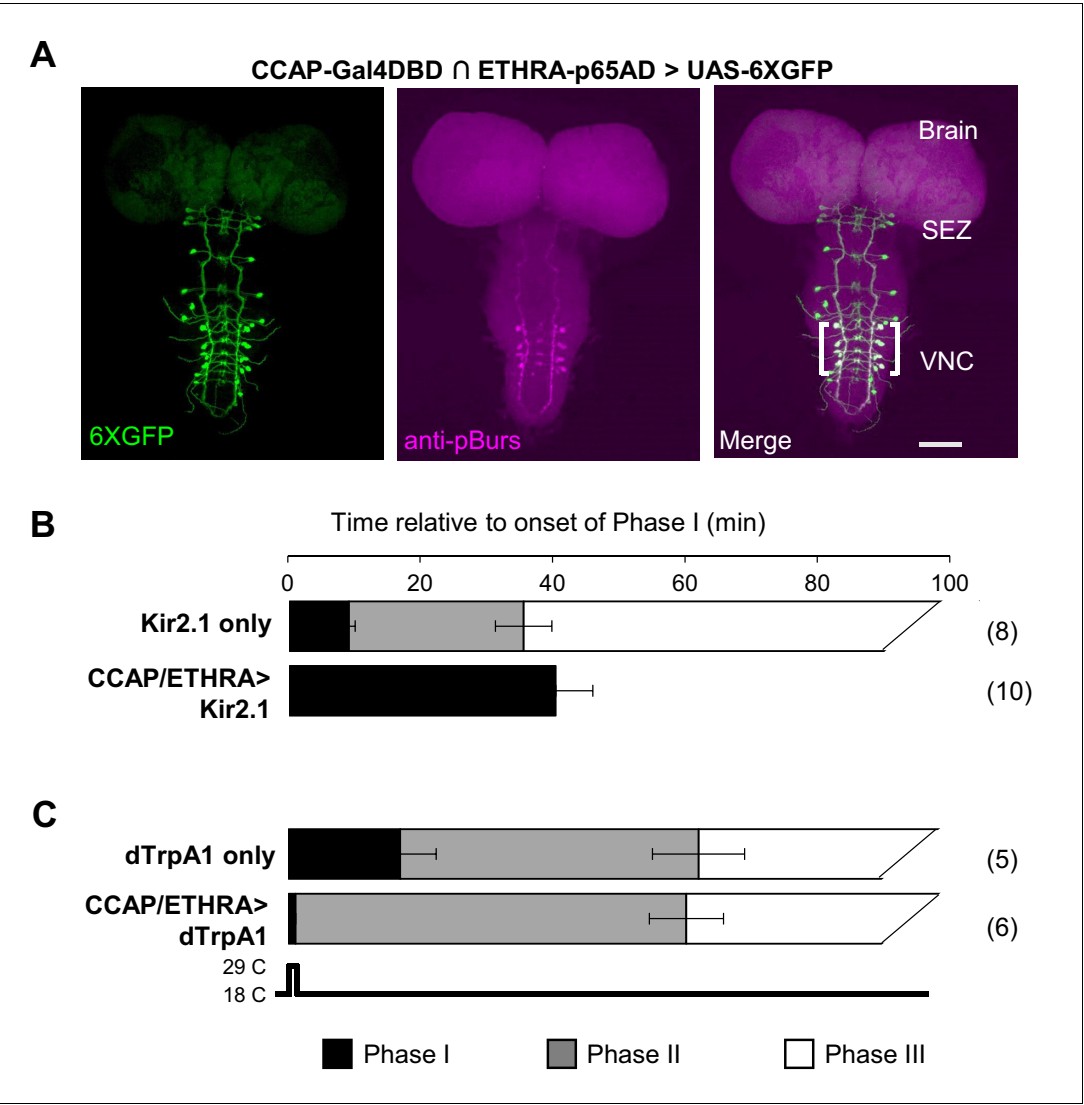

**Figure 1.** ETHRA/CCAP neuronal activity modulates all phases of pupal ecdysis. (**A**) Fluorescence confocal image of a pupal CNS wholemount. Neurons that express ETHRA, CCAP, and Bursicon are revealed by intersectional expression of UAS-6XEGFP (green, left) under the control of the CCAP-Gal4DBD∩ETHRA-p65AD hemidriver pair and anti-pBurs immunolabeling (magenta, middle). Merged image (right). Brackets, double-labeled neurons; SEZ, subesophageal zone; VNC, ventral nerve cord. Scale bar: 50 μm. (**B**) Suppression of ETHRA/CCAP neurons by two copies of UAS-Kir2.1 (bottom) eliminates both Phases II and III. Upper panel, pupal ecdysis behavior of control animals lacking the driver. Bars show the average durations of Phases I and II (±standard deviations, N in parentheses). The end of Phase III was not calculated. (**C**) Brief activation of ETHRA/CCAP neurons (1 min) using UAS-dTrpA1 terminates Phase I and initiates Phase II (bottom). Upper panel, behavior of control animals lacking the driver and subjected to the same temperature shift.

DOI: https://doi.org/10.7554/eLife.29797.005

*rickets* (*rk*) gene) or the CCAP receptor (CCAP-R). In addition, we used a previously generated and strongly expressing Rk-Gal4 driver line (Rk^pan-Gal4 *Diao and White, 2012*). The expression patterns of these lines reveal that both Rk and CCAP-R are broadly expressed in the CNS at the time of pupal ecdysis, but that few neurons express both receptors, and those only very weakly (*Figure 2A*). Rk and CCAP-R are thus expressed in almost completely distinct populations of neurons indicating that the synergistic effects of the two hormones released from the ETHRA/CCAP neurons is not due to both signals converging on a common set of targets.

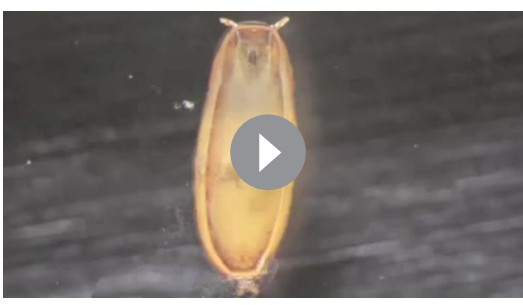

**Video 2.** Activating ETHRA/CCAP neurons terminates execution of the Phase I motor pattern and induces Phase II. ETHRA/CCAP neurons were activated using UAS-dTRPA1 by a one minute temperature shift to 29°C, followed by a return to 18°C. Video speed: 20X.
DOI: https://doi.org/10.7554/eLife.29797.004

While the effects of individually eliminating Bursicon and CCAP function by mutation differ significantly, blocking activity in the two receptor-expressing populations of neurons targeted by these factors is similar. Suppression of either population starting at the third larval instar results in severe pupal ecdysis deficits and lethality in 100% of animals (*Figure 2B*). Behavioral analysis, however, reveals small differences at the level of motor function. Animals in which the Rk-expressing neurons are suppressed are most impaired, lacking all movement and therefore all phases of the pupal ecdysis sequence (*Figure 2C*, top; *Video 3*). Interestingly, pupal development in these animals is otherwise normal: a bubble appears in the abdomen (*Figure 2B*, left, arrow) and the fat body degrades on schedule, but the overt gut movements that herald the onset of Phase I behaviors (*Robertson, 1936*) fail to appear. The presence of a sustained heartbeat throughout indicates that the animals remain viable for many hours, despite not initiating pupal ecdysis. Animals in which the CCAP-R neurons are suppressed also lack normal ecdysis behavior, executing only a mixture of weak and irregular contractions during a period of abdominal lifting with some resemblance to Phase I (*Figure 2C*, bottom; *Video 3*). We conclude that the two distinct groups of neurons targeted by Bursicon and CCAP are both essential for pupal ecdysis. We focused first on characterizing the function of the Rk-expressing neurons.

## The activity of neurons targeted by Bursicon is correlated with ecdysis motor patterns

The results of neuronal suppression demonstrate that some or all the Rk-expressing neurons are essential for initiating and/or generating all phases of the pupal ecdysis sequence. To determine whether the Rk-expressing neurons include motor neurons in the ecdysis circuit essential to its output, we performed intersectional labeling using a Rk hemidriver together with a vesicular glutamate (VGlut) hemidriver, which expresses in glutamatergic neurons, including all *Drosophila* motor neurons (*Diao et al., 2015*). We observed expression in only a handful of neurons of the thoracic ganglia, none of which extend axons to muscles, indicating that Rk is not expressed in motor neurons (*Figure 3A*). We likewise find that Rk is not highly expressed in neurons that receive the hormonal input that initiates the ecdysis sequence as determined using a Rk hemidriver in conjunction with ETHRA and ETHRB hemidrivers to identify neurons that co-express Rk and either the A- or B-isoform of the ETHR (*Figure 3A', A"*, *Diao et al., 2016*). For both isoforms, only a small number of neurons was identified that co-expressed Rk, and the suppression of these neurons with 2X UAS-Kir2.1 failed to block pupal ecdysis (data not shown). The Rk-expressing neurons essential for pupal ecdysis thus do not belong to either the input or the output layers of the ecdysis network and must therefore occupy an intermediate position in the circuit hierarchy.

To gain insight into the function of the Rk-expressing neurons, we investigated their response to upstream input from the Bursicon-expressing neurons, using the physiogenetic ATP/P2X2 system (*Yao et al., 2012*). To selectively activate the Bursicon-expressing neurons in excised pupal nervous systems by exposure to ATP, we expressed the purinergic P2X2 channel under the control of a Burs-LexA::GADfl driver and monitored the response of Rk-expressing neurons using the calcium sensor UAS-GCaMP6s driven by Rk-Gal4. $Ca^{++}$ activity was measured in the large population of neurons located in the ventral nerve cord (VNC-Rk neurons; *Figure 2A*, left, box) by laser scanning confocal microscopy, sampling at approximately 1 Hz.

We found that ATP-induced phasic $Ca^{++}$ activity in the VNC-Rk neurons, characterized by distinct, alternating, left-right oscillations across the ventral midline (*Figure 3B,C*), which were absent in control preparations lacking P2X2 expression. Quantifying the alternating oscillations, we found that the midline oscillations were, on average, sustained for 9 min. and consisted of 15 cycles of oscillation.

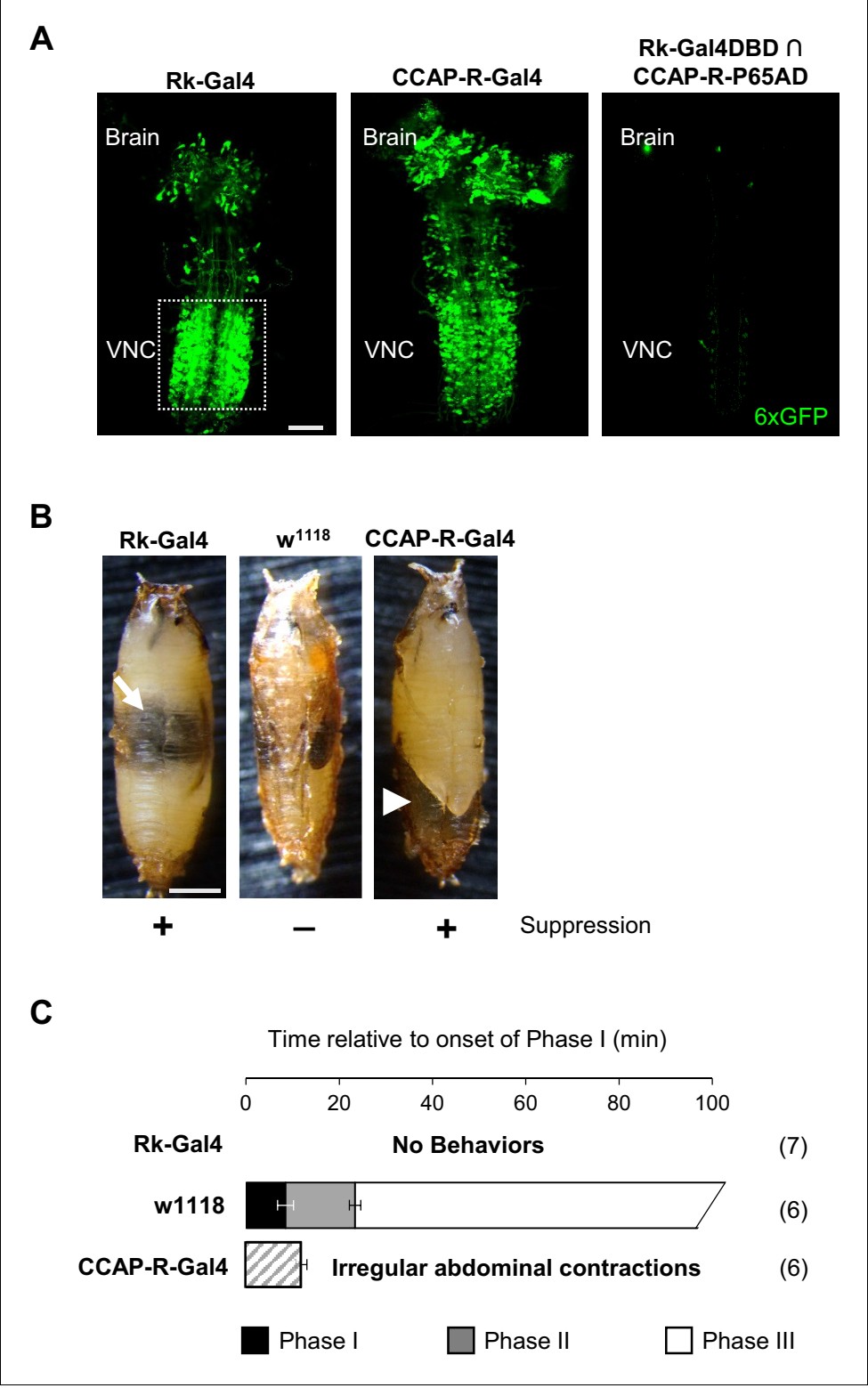

**Figure 2.** Bursicon and CCAP target distinct groups of neurons essential for pupal ecdysis. (**A**) Pupal CNS wholemounts showing neurons targeted by Bursicon (Rk-Gal4, left) and CCAP (CCAP-R-Gal4, middle). Green, UAS-6XGFP. Right panel: Intersectional labeling with Rk-Gal4DBD∩CCAP-R-p65AD hemidrivers shows that few neurons are targeted by both factors. VNC, ventral nerve cord. Scale bar: 50 μm. (**B**) Pupae in which the activity of Rk-expressing (left) or CCAP-R-expressing (right) neurons were suppressed during the period of pupal ecdysis

*Figure 2 continued on next page*

*Figure 2 continued*

using two copies of UAS-Kir2.1. Such pupae exhibit severe ecdysis deficits compared with control animals lacking a driver and therefore not subjected to neuronal suppression (middle). Arrow, air bubble abnormally retained in the body of an animal with suppressed Rk-expressing neurons. Arrowhead, large posterior air bubble in pupa with suppressed CCAP-R-expressing neurons. Scale bar: 0.5 mm. (C) Ecdysis behavior of animals subjected to suppression of the Rk-expressing neurons (top) or CCAP-R-expressing neurons (bottom) compared with control animals (middle), which execute Phase I, II, and III behaviors for the average indicated durations (±standard deviations), but animals in which Rk- or CCAP-R-expressing neurons were suppressed displayed no movements, or only disorganized contractions resembling Phase I, respectively. N, number of preparations analyzed.
DOI: https://doi.org/10.7554/eLife.29797.006

This striking pattern of activity was reminiscent of the Phase II swinging motor program, which is likewise induced by activation of neurons that express Bursicon and, as shown by muscle Ca$^{++}$ imaging below (*Figure 4C*), consists of approximately 18 bouts of alternating left-right abdominal swinging and lasts approximately 17 min. The induced activity of the Rk-expressing neurons in the isolated nervous system thus appears to be correlated with the motor output induced by a similar stimulus in the intact animal.

These results suggest that the Rk-expressing neurons may compose part of the central pattern generator governing pupal ecdysis, and because they are essential for all phases of pupal ecdysis we sought to investigate their activity throughout the ecdysis sequence. We took advantage of the fact that exposure of excised pupal nervous systems to the ETH peptide, ETH1, stimulates a fictive pupal ecdysis sequence (*Diao et al., 2016*; *Kim et al., 2006*; *Mena et al., 2016*). We reasoned that Rk-expressing neurons should be activated by ETH1 and that their temporal dynamics might reveal phasic activity corresponding to the phases of ecdysis behavior.

## ETH induces phasic activity in Rk-expressing neurons similar to the ecdysis phases

We monitored UAS-GCaMP6s activity in the VNC-Rk neurons as before and found that in excised CNS preparations treated with ETH1 these neurons clearly showed enhanced Ca$^{++}$ activity relative to preparations that did not receive ETH1 (*Figure 4A*). Compared to time traces of activity in the latter preparations, which were distinguished by relatively flat baselines and slow, low amplitude Ca$^{++}$ oscillations, the ETH1-induced traces exhibited considerable complexity. The traces could be divided into three principal phases, denoted with Arabic numbers to distinguish them from the behavioral phases, which we denote with Roman numerals. Typically, baseline Ca$^{++}$ activity rose over approximately the first 10–15 min. after ETH1 addition (*Figure 4*, Phase 1), reached a peak during the next 20 min. (Phase 2), and then slowly declined (Phase 3). Superimposed on this baseline activity were Ca$^{++}$ oscillations, which initially exhibited relatively low amplitude and high frequency, but which increased suddenly in amplitude during peak baseline activity in Phase 2. After a transition period of mixed amplitude and frequency during the baseline decline in Phase 3, the oscillations slowed and became more uniformly large and regular.

These features were sufficiently stereotyped across preparations that detection of the different Ca$^{++}$ activity phases could be automated (see Materials and methods), and using custom Matlab code (i.e. 'PhaseFinder; https://github.com/BenjaminHWhite/PhaseFinder' [*White, 2016*; copy archived at https://github.com/elifesciences-publications/PhaseFinder]; *Figure 4—figure supplement 1*) to analyze the traces, we were able to define their average times of onset (*Figure 4A*). Consistent with the

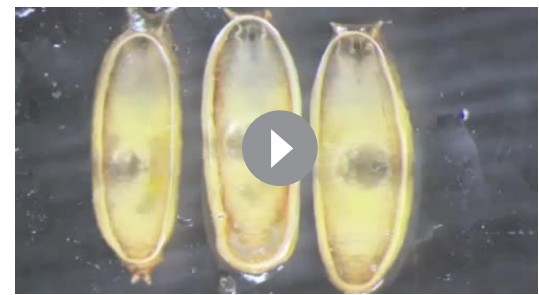

**Video 3.** Suppressing Rk- or CCAP-R-expressing neurons using UAS-Kir2.1 impairs pupal ecdysis behavior. Shown are pupae in which: CCAP-R-expressing neurons (left), Rk-expressing neurons (right), or no neurons (middle) are suppressed. Video speed: 20X.
DOI: https://doi.org/10.7554/eLife.29797.007

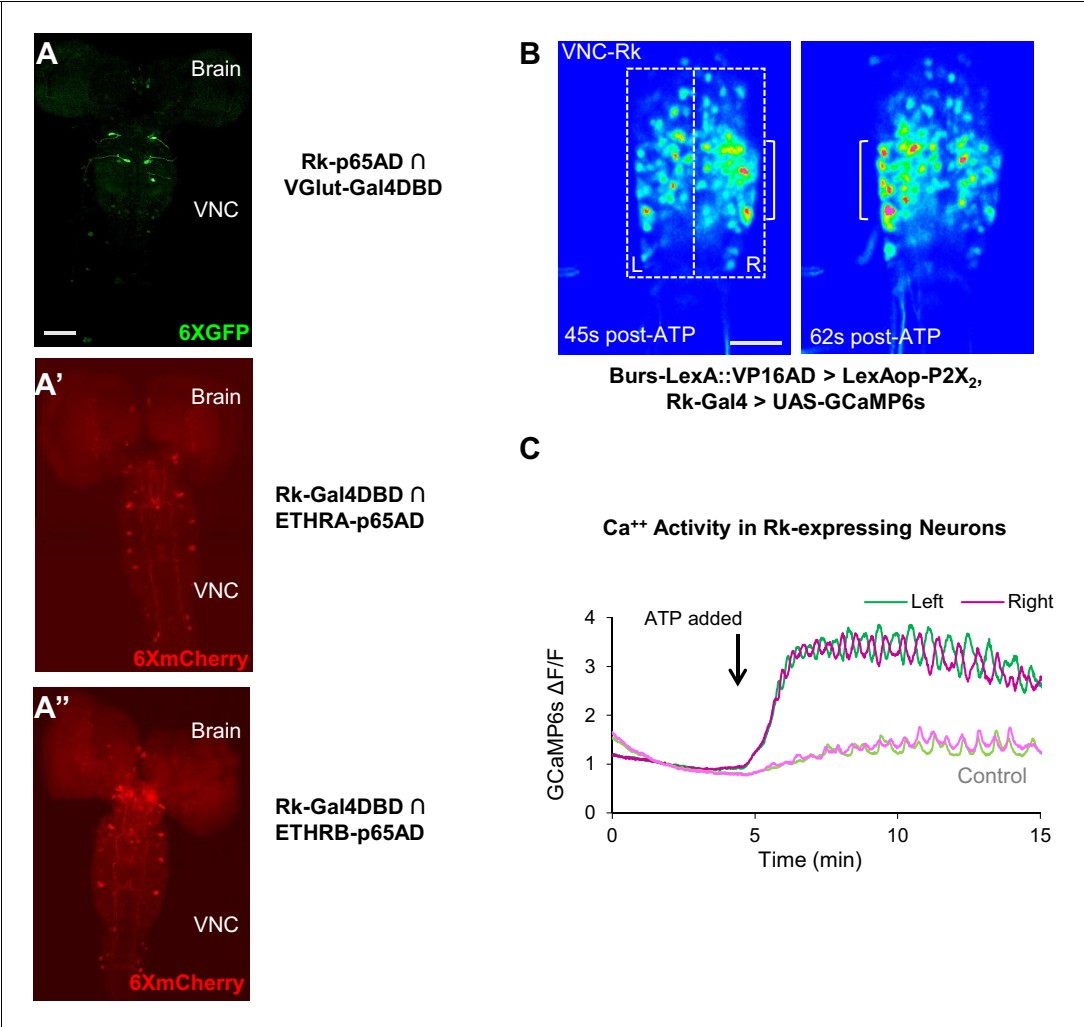

**Figure 3.** Stimulating Bursicon-expressing neurons induces Phase II-like activity in VNC-Rk neurons. (A–A") Pupal CNS wholemounts showing neurons that express Rk and either: (A) the motor neuron marker VGlut, (A') ETHRA, or (A") ETHRB, as revealed by Split Gal4 intersectional labeling. Reporters: UAS-6XGFP (green) and UAS-6XmCherry (red). VNC, ventral nerve cord. Scale bar: 50 μm. (B) Calcium activity induced in VNC-Rk neurons by activating Bursicon-expressing neurons using the purinergic P2X2 channel. ATP induces oscillatory activity in the Rk-expressing neurons, with peak $Ca^{++}$ signal alternating between the right (t = 45 s) and left (t = 62 s) sides of the ventral midline. Scale bar: 50 μm. (C) Timecourse of the GCaMP6s $Ca^{++}$ signals on the left (green) and right (magenta) side of the VNC midline (boxes in B, left panel) before and after addition of ATP (arrow). In the experimental cross (dark green and magenta), ATP induced right-left alternating peaks in the $Ca^{++}$ traces, whereas in the control cross (light green and magenta) only small, coincident oscillations characteristic of background activity were observed. Traces shown are representative of n = 6 experimental and n = 7 control preparations.

DOI: https://doi.org/10.7554/eLife.29797.008

hypothesis that activity of the VNC-Rk neurons is correlated with the phases of the ecdysis motor programs, the phases of ETH1-induced $Ca^{++}$ activity have durations similar to the ecdysis behavioral phases observed in live animals.

To permit a more direct comparison of ecdysis motor program activity with VNC-Rk neuron activity, we developed an imaging strategy that allowed us to quantify behavior by directly monitoring the $Ca^{++}$-mediated muscle contractions that drive the body wall movements (see Materials and methods). In this way, ecdysis behavior could be analyzed from $Ca^{++}$ activity signals using the same methods used to analyze the neuronal activity. To implement this strategy, we used the 24B-Gal4 driver to express UAS-GCaMP6s in muscles and monitored $Ca^{++}$ signals in animals during pupal ecdysis (*Video 4*; *Figure 4B*). The integrated $Ca^{++}$ signal over the abdominal musculature in such preparations (dotted box) typically exhibited a profile similar to that of the VNC-Rk neurons and

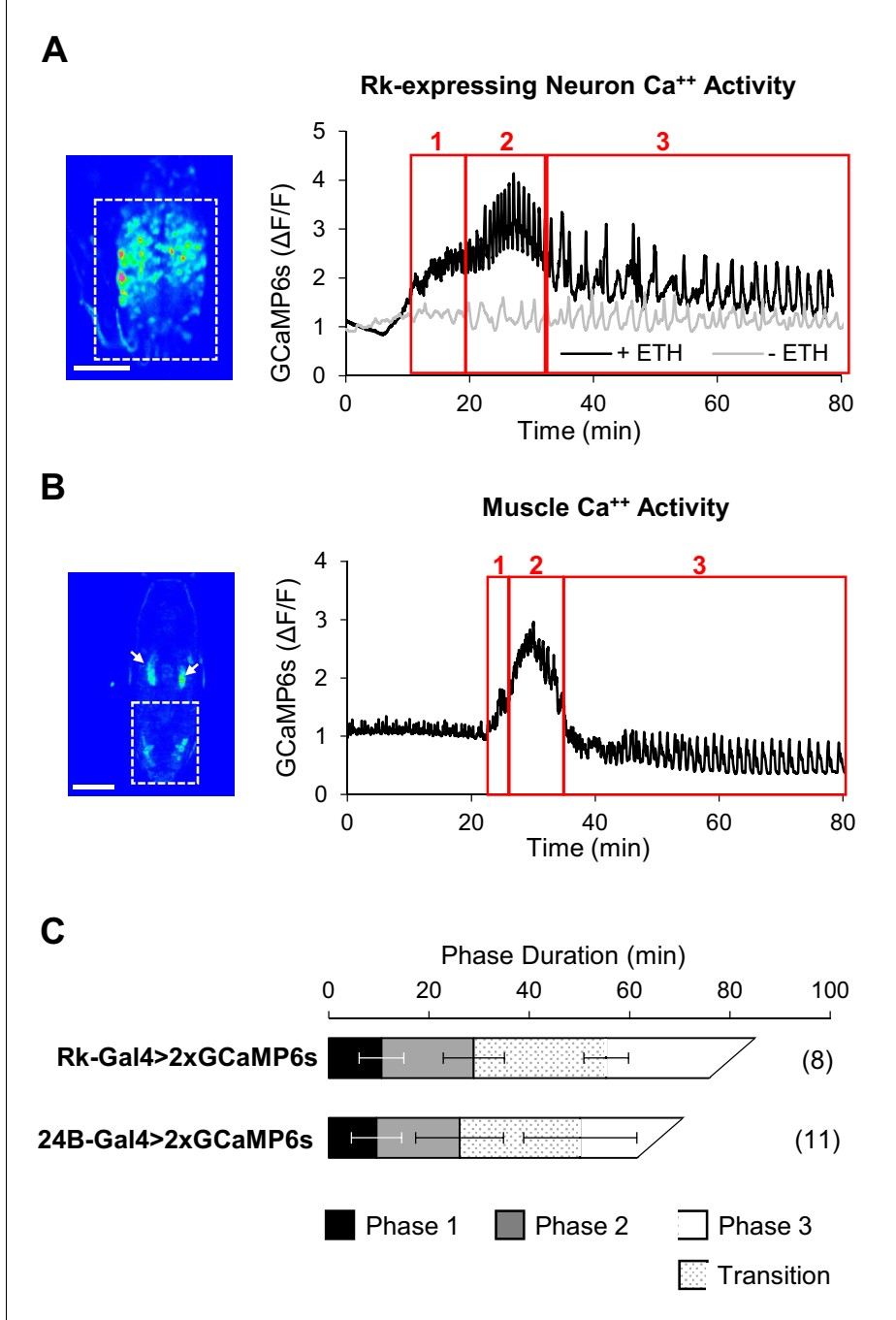

**Figure 4.** ETH1-induced Ca++ activity in Rk-expressing neurons is triphasic. (**A**) Ca++ activity in VNC-Rk neurons (measured within the dashed box, left, in an excised pupal CNS) shows a phasic response to ETH1 (black trace), distinct from the activity of control preparations not treated with ETH1 (gray trace). Three principal phases of activity could be distinguished (boxes 1, 2, and 3 as shown), the third of which displayed a transitional period with mixed activity patterns, followed by a period with more uniform oscillations. Image scale bar: 100 μm. (**B**) Ca++ activity measured in the abdominal musculature (dashed box, left) by expressing UAS-GCaMP6s under control of the 24B-Gal4 driver exhibited phases similar (boxes 1, 2, and 3) to the Rk-expressing neurons. Arrows: non-muscle salivary gland Ca++ signal. Image scale bar: 500 μm. (**C**) Ca++ phase durations in VNC-Rk neurons and abdominal muscles, as calculated by the PhaseFinder program (see Materials and methods and *Figure 4—figure supplement 1*). Both sets of traces exhibited a 'transition period' within Phase III (dotted pattern) that could also be quantified using the PhaseFinder program. Average phase durations are shown ±standard deviations; N in parentheses.

*Figure 4 continued on next page*

*Figure 4 continued*

DOI: https://doi.org/10.7554/eLife.29797.009

The following figure supplement is available for figure 4:

**Figure supplement 1.** The PhaseFinder program for automated analysis of Ca++ activity.

DOI: https://doi.org/10.7554/eLife.29797.010

divided into three principal phases when processed using the PhaseFinder program. The average onset times of these phases closely matched those calculated for the three phases of VNC-Rk activity by the same program (*Figure 4C*), again indicating a close correspondence between the neuronal activity and the three behavioral phases.

Analysis of the spatiotemporal patterns of muscle contraction visualized by Ca++ imaging also revealed details not easily seen by observation of body wall movements alone. Some details previously described only from observations of animals removed from their puparia (*Kim et al., 2006*) were clearly evident from the muscle Ca++ imaging, such as the mixed intervals of abdominal swinging and peristalsis that occur at the transition between Phases II and III. As noted above, the transition of the VNC-Rk Ca++ activity from Phase 2 to Phase 3 is also characterized by a variable interval with oscillations of mixed amplitude and frequency. To assess the similarity of this transition period to the observed interval of mixed behavior in the muscle Ca++ traces, we modified the PhaseFinder program to identify this transition period in the Phase III Ca++ data and found an analogous transition in the muscle Ca++ activity that corresponded to the interval of mixed behavior. The calculated durations of the transition periods in the two experiments were not significantly different, although both exhibited a high degree of variation across preparations (*Figure 4C*). We interpret this additional, and initially unexpected, correspondence in the VNC-Rk and muscle Ca++ data as further evidence that VNC-Rk neuronal activity is correlated with the generation of the ecdysis motor programs.

## Neurons targeted by Bursicon are responsible for central pattern generation

Further support for such a correlation comes from analyzing the spatiotemporal patterns of Ca++ signaling associated with each phase of VNC-Rk neuron activity, the characteristics of which differ from each other in ways similar to those of the ecdysis motor patterns (*Figure 5*). The similarity between the Ca++ activity of Phase 2 and the abdominal swinging of Phase II—evident in the activation experiments described above—was also seen in ETH1-induced activity data. Conspicuous left-right oscillations in the Phase 2 Ca++ signal occur both in the images (*Video 5*, *Figure 5A*) and in Ca++ traces representing the signals derived from neurons on either side of the ventral midline

(*Figure 5B*). These signals consistently oscillate in antiphase during Phase 2 (Pearson's correlation coefficient, R = −0.30 ± 0.13, p=0.0004, n = 8), but not, for example, during Phase 1, or the latter half of Phase 3 (*Figure 5C*, bottom) when the oscillations are coincident.

The corresponding analysis of muscle-generated Ca++ signals shows that Phase II is similarly characterized by strong rhythmic activity alternating across the midline with robust, temporally anti-correlated peaks (R = −0.27 ± 0.213, p=0.0016, n = 11; *Figure 5D,E*). To further compare the properties of Phase II behavior and Phase 2 Ca++ dynamics, we analyzed the frequency and number of mid-line oscillations, which for Phase II activity conforms to swings of the body wall. We found that both frequencies (0.02 ± 0.05 Hz for VNC-Rk neurons vs. 0.03 ± 0.09 Hz for 24B muscles) and oscillation

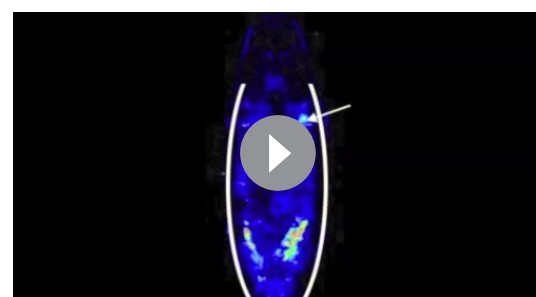

**Video 4.** Ca++ activity in body wall muscles during pupal ecdysis. GCaMP6s was expressed in muscle using the 24B-Gal4 driver. Solid line indicates boundary of the pupal case below the head. A non-muscle, ETH-induced signal in the salivary glands is also visible. Video record: total time, 45 min collected at 1 Hz; video speed: 20X.

DOI: https://doi.org/10.7554/eLife.29797.011

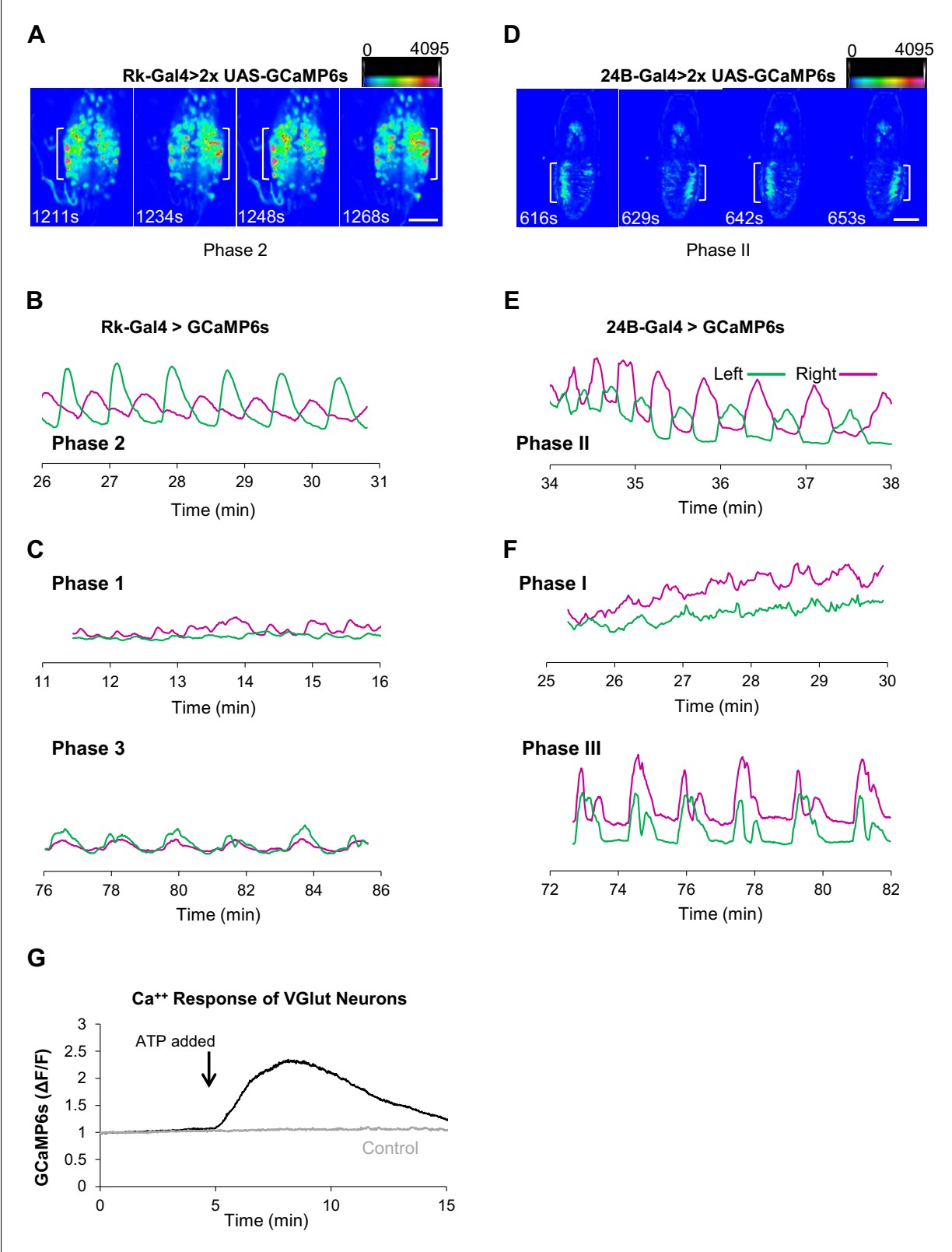

**Figure 5.** Rk-expressing neurons act in central pattern generation. (**A–C**) Analysis of ETH1-induced Ca$^{++}$ activity in VNC-Rk neurons. (**A**) Images from two complete cycles of alternating Ca$^{++}$ signal in VNC-Rk neurons during Phase 2. Images correspond to the indicated times of peak signal (brackets) on each side of the midline. Scale bar: 100 µm. (**B**) Phase 2 Ca$^{++}$ signals from the preparation shown in (**A**) measured on the right (magenta) and left (green) sides of the midline (as shown in **Figure 3B**) oscillate in anti-phase. (**C**) Ca$^{++}$ signals corresponding to Phase 1 (top) measured on either side of

*Figure 5 continued on next page*

*Figure 5 continued*

the midline are poorly correlated, whereas those corresponding to Phase 3 (bottom) are in phase. (For more detailed analysis of Ca$^{++}$ signals from selected small ROIs within the VNC-Rk neurons see *Figure 5—figure supplement 1*.). (D–F) Analysis of Ca$^{++}$ activity in abdominal muscles during the pupal ecdysis sequence. (D) Images from two complete cycles of alternating Ca$^{++}$ signal in the abdominal musculature during Phase 2. Times and brackets indicate peak signal on each side of the midline, as in (B). Scale bar: 500 µm. (E) Phase 2 Ca$^{++}$ signals measured on the right (magenta) and left (green) sides of the midline. (F) Ca$^{++}$ signals corresponding to Phase 1 (top) measured on either side of the midline are poorly correlated, whereas those corresponding to Phase 3 (bottom) are in phase. (G) Timecourse of Ca$^{++}$ activity in VGlut-expressing (motor) neurons of the abdominal ganglia in preparations in which the Rk-expressing neurons were activated (black trace) by ATP (arrow), or not activated because the P2X2 channel was not expressed (gray trace). Traces shown are representative of n = 6 experimental and n = 6 control preparations.

DOI: https://doi.org/10.7554/eLife.29797.013

The following figure supplement is available for figure 5:

**Figure supplement 1.** VNC-Rk responses to ETH1 differ.

DOI: https://doi.org/10.7554/eLife.29797.014

numbers (23 ± 7 for VNC-Rk neurons vs. 18 ± 2 for 24B muscles) were similar. Taken together, these data are consistent with the hypothesis that Phase 2 activity of the VNC-Rk neurons drives the execution of the Phase II motor program.

There is also evidence that the VNC-Rk neuron activity of Phases 1 and 3 similarly generates the motor patterns of Phases I and III of pupal ecdysis. We have already noted that the Transition Period of Phase 3 has an apparent behavioral correlate in the mixed Phase II and III behaviors observed in the muscular activity. In addition, the muscular activity responsible for the stretch compressions of Phase III exhibit bilaterally coincident peaks of Ca$^{++}$ activity (R = 0.57 ± 0.14, p<0.0001, n = 11; *Figure 5F*, bottom traces), similar to those of VNC-Rk neuron activity during late Phase 3 (*Figure 5C*, bottom). In some preparations, symmetric and rhythmic anterior-to-posterior waves of VNC-Rk neuron Ca$^{++}$ activity are also evident, as might be expected for the neuronal activity that drives Phase III abdominal peristalsis (*Video 6*). Similarly, the distinct spatiotemporal patterns of VNC-Rk neuron activity associated with Phase 1 are frequently reminiscent of the lateralized, alternating posterior-to-anterior peristaltic waves of contraction that traverse the body wall during Phase I (*Video 7*). These patterns, however, differ from those of Phase 2 in that they are not generated by anatomically isolated neuronal populations that dominate the global Ca$^{++}$ signal, but instead derive from multiple, anatomically intermingled signals.

Resolving the individual components of these intermingled signals and reproducibly identifying them across preparations will require more refined methods, but to perform a preliminary decomposition of the Ca$^{++}$ signal produced by the Rk-expressing neurons, we analyzed the activity in a representative preparation, examining 95 small regions of interest (ROIs; *Figure 5—figure supplement 1A,B*). Although we cannot be certain that these ROIs correspond to individual cells due to the limited resolution of the Ca$^{++}$ signal in the z-dimension, their Ca$^{++}$ activity traces fell into two broad categories: those with large, slow changes in baseline amplitude (i.e. #28, 84, 16), and those with oscillatory activity, but relatively constant baseline (i.e. #89, 2, 74). This suggests that the VNC-Rk population contains at least two types of neurons: one that may represent the oscillatory output of the network and one that may be involved in sustaining phasic activity within it. Records for both types of ROI, contained examples in which activity was predominantly restricted to one (#28, 89), two (#84, 2), or three (#16, 74) phases and some records with oscillatory activity exhibited changes in oscillation frequency with phase (e.g. #2, 74). Analysis of the average frequency of oscillations as a function of phase for all 95 ROIs indicates that approximately 10% of the ROIs had frequency profiles similar to that of the global Ca$^{++}$ trace (*Figure 5—figure supplement 1C*). This observation is consistent with the conclusion that

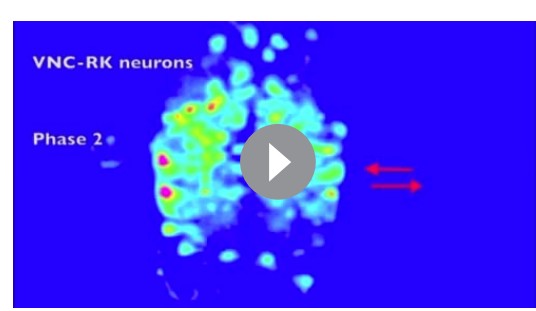

**Video 5.** ETH-induced activity in VNC-Rk neurons oscillates across the midline during Phase 2. Video record: collected at 1 Hz; video speed: 50X.

DOI: https://doi.org/10.7554/eLife.29797.012

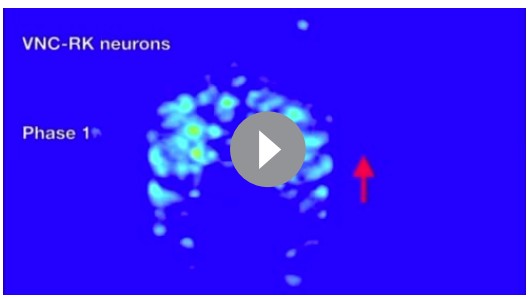

**Video 6.** ETH-induced activity in VNC-Rk neurons during Phase 3. Video record: collected at 1 Hz; video speed: 50X.
DOI: https://doi.org/10.7554/eLife.29797.015

**Video 7.** ETH-induced activity in VNC-Rk neurons during Phase 1. Video record: total time 90 min, collected at 1 Hz; video speed: 50X.
DOI: https://doi.org/10.7554/eLife.29797.016

multifunctional neurons participate in the generation of all three phases of behavior, but more refined observation and manipulation of individual VNC-Rk neurons will be required to definitively determine their role(s) in central pattern generation. Overall, however, our analysis of the spatiotemporal structure of ETH1-induced $Ca^{++}$ activity of the Rk-expressing neurons suggests that they comprise a multifunctional central pattern generator (CPG) for the pupal ecdysis sequence.

## CCAP-R-expressing neurons include motor neurons that generate the ecdysis rhythms

The hypothesis that Rk-expressing neurons act as central pattern generators for the ecdysis rhythms predicts that these neurons communicate their output to downstream motor neurons. To directly determine whether motor neurons receive input from Rk-expressing neurons, we used the ATP/P2X2 system to stimulate Rk-expressing neurons while monitoring the response of glutamatergic neurons in the VNC—95% of which are motor neurons (*Daniels et al., 2008*). To do so, we used the VGlut-LexA::QFAD driver to express LexAop-GCaMP6s, while expressing UAS-P2X2 under the control of Rk-Gal4. We found that ATP induced a substantial increase in $Ca^{++}$ signal in a large number of neurons (*Figure 5G*), indicating that glutamatergic motor neurons of the VNC are downstream targets of the Rk-expressing neurons.

The identity of some or all the motor neurons activated by the stimulation of Rk-expressing neurons was revealed by our parallel investigation of the targets of CCAP signaling. As shown above, CCAP-R-expressing neurons are essential for pupal ecdysis, and as for the Rk-expressing neurons, we used intersectional methods to ask whether they include ETHR-expressing neurons of the input layer or glutamatergic motor neurons of the output layer. We find that very few CCAP-R-expressing neurons in the pupal CNS co-express either of the ETHR isoforms, but that a significant complement are glutamatergic, as identified by the intersectional driver for CCAP-R and VGlut. A fillet preparation of the pupal body wall in which the muscles are labeled with phalloidin (*Figure 6A*, magenta), reveals that many of the glutamatergic neurons labeled by UAS-6XGFP (*Figure 6A*, green), are motor neurons, sending their axons out the abdominal nerves and forming synapses on muscles in the body wall (*Figure 6A'*, arrowheads).

Using the same intersectional driver to express UAS-GCaMP6s in these neurons (i.e. CCAP-R/VGlut neurons), we selectively monitored their response to ETH1 in excised pupal CNS preparations. Interestingly, the profile of the induced $Ca^{++}$ activity is multiphasic and decomposes into three principal phases as distinguished by the PhaseFinder program (*Figure 6B*). As for the $Ca^{++}$ activity of the VNC-Rk neurons, the last phase contains an identifiable transition period with $Ca^{++}$ oscillations of mixed amplitude and frequency, and the activity of Phase 2, when analyzed bilaterally across the midline, shows characteristic left-right alternations (*Figure 6C*). The durations (*Figure 6D*) and frequencies (*Figure 6E*) of $Ca^{++}$ activity for the CCAP-R/VGlut and VNC-Rk neurons were comparable, with no statistically significant differences found for any Phase. These observations are consistent with a tight coupling between the activities of the VNC-Rk and CCAP-R/VGlut motor neurons. To directly determine whether activity in CCAP-R neurons can be driven by the Rk-expressing neurons, we used the ATP/P2X2 system and found that selective activation of the Rk-expressing neurons

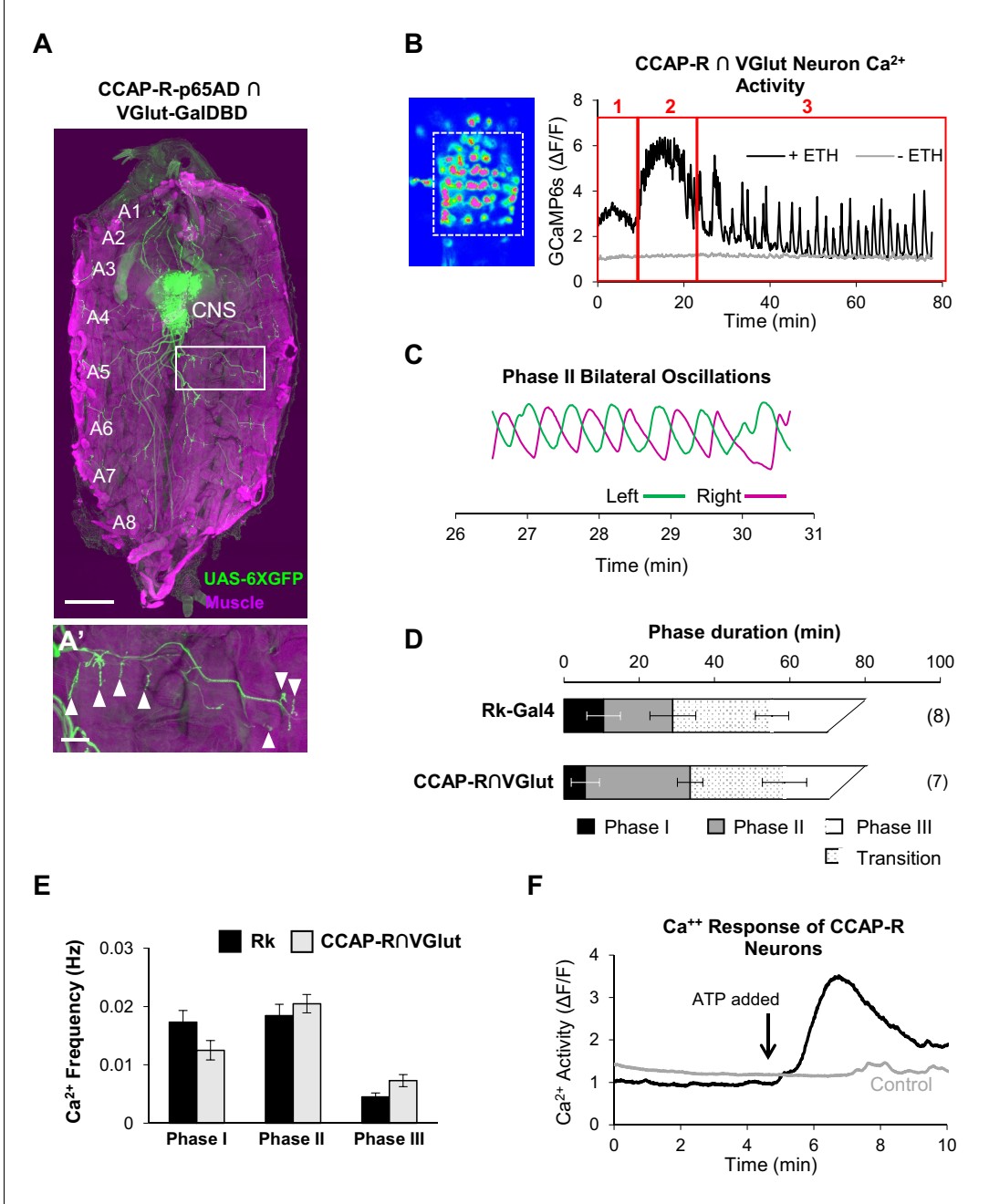

**Figure 6.** CCAP-R-expressing motor neurons act downstream of Rk-expressing neurons. (A, A′) A late 3rd instar larval fillet in which body wall muscles are stained with phalloidin (magenta) and a UAS-6XGFP reporter (green) reveals the expression pattern of a CCAP-R-p65AD∩VGlut-GalDBD Split Gal4 driver. Glutamatergic motor neurons within this pattern project their axons to muscles. Scale bar: 0.5 mm. (A′) Axons in one hemisegment (white box in A) can be seen to terminate in neuromuscular synapses (arrowheads). Scale bar: 100 µm. (B) Ca$^{++}$ activity in CCAP-R/VGlut neurons (measured within the boxed area on left from the VNC of an excised pupal CNS) shows a phasic response to ETH1 (black trace), distinct from the activity of control preparations not treated with ETH1 (gray trace). Three principal phases of activity can be distinguished (red boxes 1, 2, and 3). (C) Ca$^{++}$ signals from Phase 2 measured on the right (magenta) and left (green) sides of the midline oscillate in anti-phase. (D) Durations of Ca$^{++}$ activity phases in VNC-Rk neurons and CCAP-R/VGlut neurons, as calculated by the PhaseFinder program. (Error bars show standard deviations; N in parentheses.) Both sets of traces exhibited a 'transition period' within Phase III (dotted pattern) that could also be quantified using the PhaseFinder program. (E) Average frequency of Ca$^{++}$ oscillations observed in VNC-Rk neurons (black) and CCAP-R/VGlut neurons (gray) for the three phases of Ca$^{++}$ activity. For Phase III, only the frequency of the uniform activity after the end of the transition period was calculated. Error bars show standard deviations. The frequencies of the two sets of neurons did not differ significantly for any Phase, when compared by ANOVA. (As shown in *Figure 6—figure supplement 1*, the neurons that express CCAP-R and Rk are largely distinct.). (F) Time course of Ca$^{++}$ activity in CCAP-R neurons of the abdominal ganglia in preparations

*Figure 6 continued on next page*

*Figure 6 continued*

in which the Rk-expressing neurons were activated (black trace) by ATP (arrow), or not activated by ATP because the Rk$^{TGEM}$-LexA::QFAD driver was omitted (gray trace). Traces shown are representative of n = 6 experimental and n = 7 control preparations.

DOI: https://doi.org/10.7554/eLife.29797.017

The following source data and figure supplement are available for figure 6:

**Source data 1.** Ca++ Oscillation Frequencies for Rk and CCAP-R Motor Neurons.

DOI: https://doi.org/10.7554/eLife.29797.019

**Figure supplement 1.** CCAP-R- and ETHR-expressing neurons are largely distinct.

DOI: https://doi.org/10.7554/eLife.29797.018

induced a large Ca$^{++}$ response in the CCAP-R VNC neurons (*Figure 6F*). We conclude that motor neurons expressing the CCAP-R act downstream of CPG neurons that express the Bursicon receptor, Rickets.

## Regulation of Phase I requires non-CCAP, non-Bursicon inputs

The above data are consistent with a model in which the hormones Bursicon and CCAP, released from neurons in the input layer of the ecdysis network, act on two subsequent layers in the network hierarchy responsible for central pattern generation and motor output. The latter two layers, defined by their expression of Bursicon- and CCAP-receptors, respectively, appear to be broadly involved in generating all three phases of the ecdysis sequence. Bursicon and CCAP, however, are unlikely to be released from the ETHRA/CCAP neurons until Phase II, the first motor program for which they are required, and their regulation of Phase I behavior is inhibitory. The initiation and maintenance of Phase I is thus likely to depend on other factors. To ask which neurons within the input layer might regulate Phase I, we used two drivers that together allowed us to interrogate the function of the two subsets of ETHR-expressing neurons outside of those that secrete CCAP and Bursicon. These drivers target neurons that express the B-isoform of the ETHR (i.e. ETHRB-Gal4; *Figure 7A*), a population almost completely distinct from the ETHRA-expressing subset (*Diao et al., 2016*), and neurons that express ETHRA, but do not co-express CCAP (i.e. non-CCAP/ETHRA neurons; *Figure 7B*).

Silencing the activity of ETHRB-expressing neurons with UAS-Kir2.1, caused a variety of behavioral defects, including rapid, tremulous movements of the body wall, but the most salient feature was the lack of an overt Phase I (*Figure 7C*; *Video 8*). Phase II and Phase III motor programs were readily discernible in affected animals, but the abdominal lifting which initiates Phase I was absent, and although some abdominal movements preceded Phase II, they appeared to be exaggerated versions of the body wall movements that normally prefigure the onset of Phase I, and they never displaced the abdominal air bubble as they do in normal animals.

Animals in which the non-CCAP/ETHRA neurons were silenced also exhibited obvious deficits in Phase I, which was executed by most animals (n = 10/15), but was consistently shorter than it was in controls (3.82 ± 2.8 vs. 7.42 ± 3.5, p=0.0137; *Figure 7D*; *Video 9*). This behavioral difference was not attributable to the bloating commonly observed in these animals due to suppression of ETHRA-expressing neurons that co-express the diuretic hormone Leucokinin (*Diao et al., 2016*). Bloating did, however, make it difficult to consistently distinguish Phases II and III, which sometimes appeared persistently intermingled. These results indicate specific requirements for ETHRB and non-CCAP/ETHRA neurons in the initiation and maintenance of Phase I. Interestingly, Phase I was not induced by activating either of these populations of neurons prior to the onset of ecdysis using UAS-TrpA1, and further experiments will be required to identify the sufficient causes for initiation of this phase.

## Discussion

Using the Trojan exon method to selectively target populations of hormone receptor-expressing neurons for manipulation and monitoring of activity, we have investigated the neuromodulatory connectivity of the circuitry governing pupal ecdysis behavior in *Drosophila*. We find that the sites of action of the neuromodulators ETH, Bursicon, and CCAP identify essential functional components of the network architecture, defining three hierarchically organized layers from the sites of hormonal initiation to the sites of motor neuron output. In addition, we find that descending neuromodulatory signaling from the ETHR-expressing input layer not only governs the basic motor rhythms of the

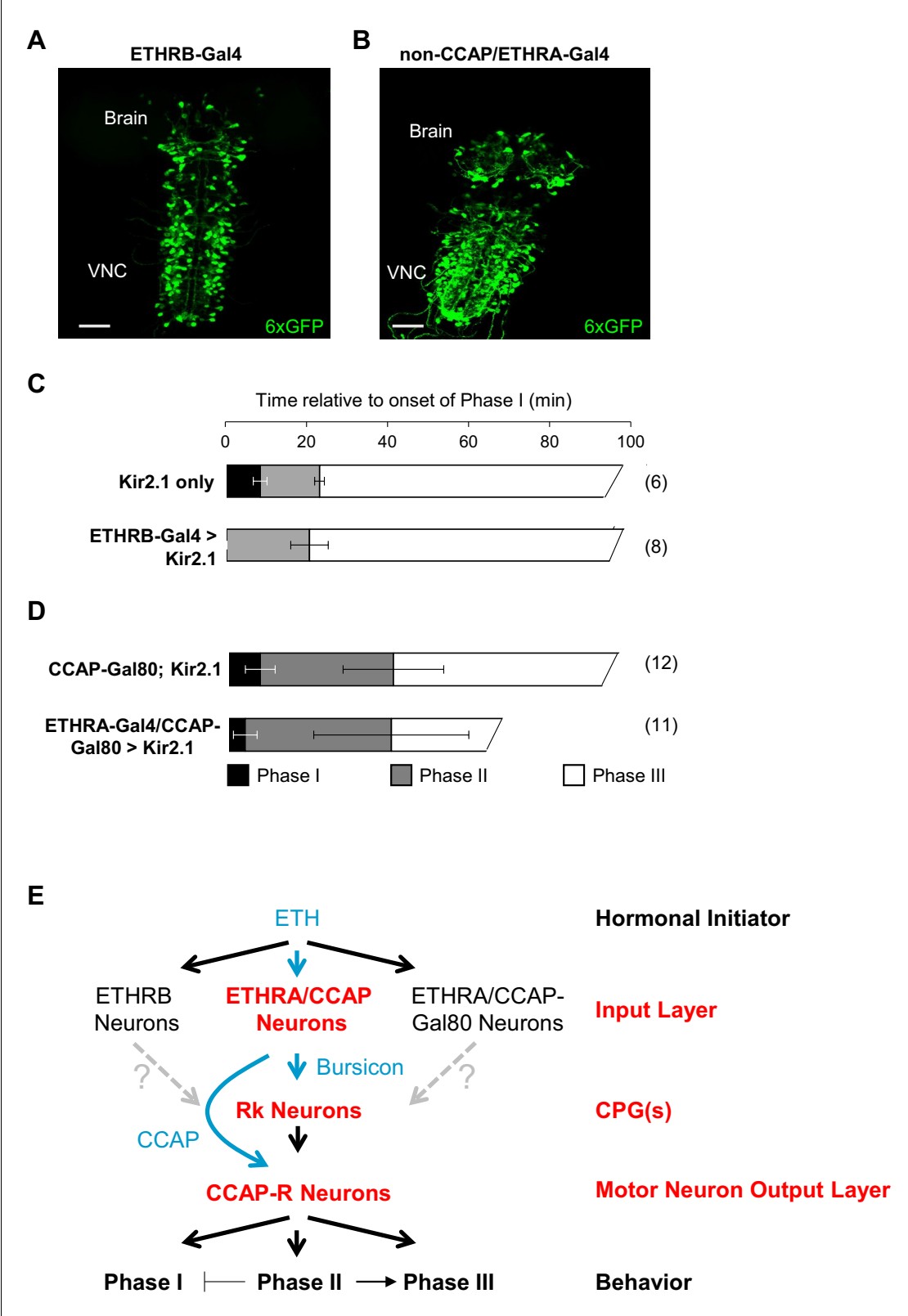

**Figure 7.** ETHRB-expressing and non-CCAP/ETHRA neurons regulate Phase I. (**A**) Pupal CNS wholemount showing the expression pattern of ETHRB-Gal4 (green, UAS-6XGFP). VNC, ventral nerve cord. Scale bar: 50 μm. (**B**) Similar to (**A**), but showing the expression pattern of ETHRA-Gal4 excluding the neurons that express CCAP. Gal4 activity in the latter neurons was blocked using CCAP-Gal80. Scale bar: 50 μm. (**C–D**) Behavioral effects of suppressing either: (**C**) ETHRB-expressing neurons, or (**D**) non-CCAP/ETHRA neurons. Unlike unsuppressed control animals (upper graphs), animals in

*Figure 7 continued on next page*

*Figure 7 continued*

which ETHRB-expressing neurons are suppressed (C, bottom) lack Phase I, and this phase is significantly shortened (p-0.014) by suppression of non-ETHRA/CCAP neurons (D, bottom). Bar graphs show average phase durations ± standard deviations, (N in parentheses). (**E**) Hierarchical organization of the pupal ecdysis circuit. Each layer in the network hierarchy (red) is the target of one of the ecdysis hormones (blue), as defined by expression of its receptor. The most important component of the input layer, ETHRA/CCAP, participates in regulating all three behavioral phases and is the source of CCAP and Bursicon. Bursicon's receptor, Rk, defines a central pattern generating layer, which sends output to a population of motor neurons that express the receptor for CCAP and are required for the ecdysis sequence. Solid arrows indicate demonstrated functional connections, while gray, dashed arrows indicate hypothesized connections. Although the detailed mechanisms governing motor program progression remain to be determined, those that promote Phase II negatively regulate Phase I and positively regulate Phase III.

DOI: https://doi.org/10.7554/eLife.29797.020

ecdysis sequence by modulating the intermediate CPG layer, but also modulates activity of the CCAP-R-expressing motor neurons of the output layer (*Figure 7D*). Neuromodulators thus act broadly within, as well as across, network layers. Our finding that the functional architecture of the ecdysis network can be decoded from its patterns of neuromodulatory connectivity provides further evidence that characterizing neuromodulatory connectomes is a valuable strategy in elucidating neural networks.

## Major components of the pupal ecdysis circuitry are shared by the three motor programs

The schematic shown in *Figure 7D* broadly augments existing models of the pupal ecdysis network (*Kim et al., 2015*; *Mena et al., 2016*; *Zitnan and Adams, 2012*), both by providing a more comprehensive description of the input layer than has previously been possible and by identifying the motor circuits on which this layer acts. A principal finding reported here is that the downstream targets of Bursicon and CCAP are shared components of the pupal ecdysis network and are used to generate all three motor rhythms. Our results draw particular attention to the centrality of neurons that express the Bursicon receptor (Rk), which are absolutely required for all pupal ecdysis behavior. A role in central pattern generation is indicated both from the effects of their suppression, which eliminates all motor activity, and from their pattern of ETH1-induced $Ca^{++}$ activity, which matches the phases of ecdysis behavior. The fact that ETH1-induced $Ca^{++}$ activity is observed in the excised nervous system and thus in the absence of sensory feedback, demonstrates that it is centrally generated and further supports the identification of the VNC-Rk neurons as central pattern generators. Conclusive evidence that some or all VNC-Rk neurons participate in central pattern generation will require more precise observations and perturbations than those performed here, as will determining the functional roles of individual neurons. However, our preliminary observation that regions containing at most small numbers of VNC-Rk neurons exhibit activity that is phasically coupled to two or more motor patterns argues that the ecdysis circuitry includes multifunctional CPG neurons that express Rk and are subject to modulation by distinct input layer modules, as indicated in *Figure 7D*. Similar architectures have been described in other motor networks where two CPGs formed from overlapping pools of neurons can switch between activity states to generate distinct behaviors (*Kristan and Gillette, 2007*).

## Input layer control of the phases of pupal ecdysis

How input layer neurons modulate the pupal ecdysis CPG is exemplified by the control of Phase II by ETHRA/CCAP neurons. Direct activation of these neurons induces Phase II-like rhythmic activity in the VNC-Rk neurons, an observation that is easily explained if Bursicon secreted from ETHRA/CCAP neurons shifts the mode of activity of the VNC-Rk CPG. This mechanism is consistent with the neuromodulatory

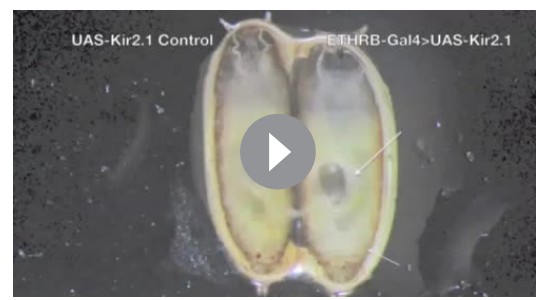

**Video 8.** Suppressing ETHRB expressing neurons using UAS-Kir2.1 eliminates pre-ecdysis behavior. Right: pupa in which ETHRB-expressing neurons are suppressed. Left: unsuppressed control animal. Video speed: 30X
DOI: https://doi.org/10.7554/eLife.29797.021

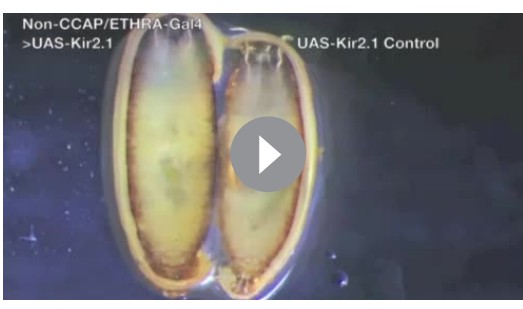

**Video 9.** Suppressing non-CCAP/ETHRA neurons shortens Phase I behavior. Left: pupa in which non-CCAP/ETHRA neurons are suppressed. Right: unsuppressed control animal. Video speed: 30X.
DOI: https://doi.org/10.7554/eLife.29797.022

control of CPGs described in numerous other systems (*Briggman and Kristan, 2008*; *Dickinson, 2006*; *Marder and Bucher, 2007*) and accounts for the long-standing observation that CCAP- and Bursicon-expressing neurons are important for pupal ecdysis (*Kim et al., 2006*; *Lahr et al., 2012*; *Park et al., 2003a*; *2008*), including Phase II initiation and Phase I termination (*Kim et al., 2015*). The CCAP- and Bursicon-expressing neurons are known to express additional neuropeptides, including Myoinhibitory Peptides and Allatostatin C, and it is likely that these neuromodulators also play a role in regulating these phases. The mixed activity patterns that define the transition from Phase II to Phase III, first described by *Kim et al. (2006)* and further characterized here, are also readily interpreted as a period of bistability in which CPG modes transiently alternate, perhaps as Bursicon and/or other co-released neuromodulator concentrations fall.

In addition to neurons that switch CPG activity from Phase I to Phase II, the input layer must also contain neurons that initiate pupal ecdysis by inducing Phase I. The search for such neurons has focused primarily on those that express ETHRA (*Kim et al., 2006*; *Krüger et al., 2015*; *Lahr et al., 2012*; *Mena et al., 2016*), but no components of this group have yet been identified that are required for ecdysis initiation. To identify the ETH targets responsible for Phase I, we systematically parsed ETHR-expressing neurons into three, nearly mutually exclusive subsets that together cover the entire input layer. Our results indicate that the largely uncharacterized neurons that express the B-isoform of ETHR are required to initiate Phase I, and that the non-CCAP/ETHRA neurons are important for maintaining that phase.

The essential role of ETHRB-expressing neurons in Phase I initiation is consistent with the significantly higher affinity for ETH peptides of ETHRB compared with ETHRA (*Iversen et al., 2002*; *Park et al., 2003b*). ETHRB-expressing neurons may thus initiate Phase I by responding to rising titers of ETH earlier than neurons expressing ETHRA. How they regulate the VNC-Rk CPG neurons remains to be determined, but their mechanism of action appears to be different from that of the ETHRA/CCAP neurons insofar as the Phase I motor program cannot be evoked by TrpA1-mediated activation. It could be that this manipulation fails to induce the correct pattern of activity in ETHRB-expressing neurons. Preliminary imaging results show that ETHRB-expressing neurons respond to ETH1 with oscillatory activity (data not shown), and it is possible that these neurons directly couple to the Rk-expressing neurons through synaptic or electrical contacts and participate in generating Phase I behavior. However, further characterization of the activity of both the ETHRB- and non-CCAP/ETHRA neurons will be required to determine how they modulate VNC-Rk CPG activity.

Two input layer neurons that are common to the ETHRB- and non-CCAP/ETHRA groups express the major ecdysis neuromodulator, EH (*Diao et al., 2015*). The EH-expressing neurons, which are among the few cells to express both ETHRA and ETHRB, respond to ETH1 application at the onset of Phase II (*Kim et al., 2006*), and evidence from other insects indicates that EH targets CCAP-expressing neurons (*Ewer and Truman, 1996*). EH is thus thought to be responsible for the release of CCAP and Bursicon, but this has not yet been verified in *Drosophila* where the EH receptor has yet to be identified. We were thus not able to target EH receptor-expressing neurons in this study, but the identity and function of such neurons is likely to be critical to understanding the progression of the ecdysis sequence.

In general, it is worth noting that the neuromodulators regulating the ecdysis sequence are of the type called 'extrinsic,' because they are released from neurons that do not function in the circuits upon which they act (*Katz and Frost, 1996*). Extrinsic neuromodulatory neurons, however, must be components of the broader neural networks that generate behaviors, and the mechanisms that organize their activities are only beginning to be understood (*Brezina, 2010*). In some cases, these mechanisms are surprising. For example, the neuromodulatory connections between neurons that govern two foraging states in *C. elegans* are orthogonal to the sensory-to-motor synaptic connections

between these neurons, which are not involved in the state decision (*Flavell et al., 2013*). There are currently few studies that jointly examine patterns of neuromodulatory and synaptic connectivity (but see *Schlegel et al., 2016*), and to understand how extrinsic neuromodulatory neurons integrate into the broader networks in which they function more examples of such networks are required. Elucidating the interactions of neurons in the input layer of the ecdysis network—in addition to interactions of the input layer with neurons in other layers—should provide insight into this general problem.

## Feed-forward mechanisms in the regulation of motor output: the role of CCAP

Our finding that the motor output of the pupal ecdysis network is mediated by neurons that express the CCAP-R provides insight into the hitherto poorly understood mechanism of action of CCAP. This neuropeptide plays critical roles in the ecdysis of other insects (*Arakane et al., 2008*; *Gammie and Truman, 1997*), but genetic data demonstrate that in *Drosophila* it plays a subsidiary role to Bursicon, acting synergistically with that hormone to render pupal ecdysis more robust (*Lahr et al., 2012*). Our results indicate that it does so by acting on motor neurons, and because CCAP is co-released with Bursicon from the ETHRA/CCAP neurons to govern the CPG transition at Phase II, this suggests a role for feed-forward signaling in the pupal ecdysis circuit.

Neuromodulatory feedforward pathways have been previously described (*Wu et al., 2010*) and appear to be a common motif in motor network architectures (*Taghert and Nitabach, 2012*). Feed-forward loops of the type posited here for Bursicon and CCAP may be important in adjusting the coupling between Rk-expressing CPG neurons and their downstream motor neuron targets during Phase II. Compensatory adjustments in CPG, motor neuron and muscle activity by a single neuro-peptide released from two different nodes in a feedforward loop have been described in the *Aplysia* feeding network where they guarantee stability of network output (*Jing et al., 2010*). Coordinating CPG activity with motor neuron activity may be particularly important for multifunctional CPGs, in which individual neurons participate in multiple motor patterns, as for example, in the leech swim/crawl network in which multifunctional neurons fire in phase with the contraction of one muscle group during swimming, but not necessarily during crawling (*Briggman and Kristan, 2006*).

## The value of 'neuromodulatory connectomics'

The architecture of the pupal ecdysis network revealed here is similar to that of other motor circuits, such as those governing locomotion, feeding, and breathing in which higher order neurons modulate the activity of core CPGs to generate varied motor patterns (*Feldman et al., 2013*; *Mullins et al., 2011*; *Nusbaum et al., 2001*). What is striking about neuromodulator action in the ecdysis circuit is its broad scope. ETH acts throughout the input layer to control different phases of pupal ecdysis behavior; Bursicon similarly regulates a large and essential set of neurons constituting the ecdysis CPG; and CCAP acts on many motor neurons necessary for generating the rhythmic ecdysis movements. The observation that Bursicon and CCAP signal from the input layer speaks to an organizational logic in which the ecdysial neuromodulators function together to provide coherence to the operation of the pupal ecdysis network by acting both within each hierarchical layer and by acting coordinately across layers. This organization is consistent with a generalized role for neuromodulatory systems in organizing neural activity to generate behavior (*Marder, 2012*).

Our results also support the rationale of mapping neuromodulatory pathways as a strategy for identifying essential network circuits and their functional organization. It is worth noting that our mapping of the pupal ecdysis network was done without reference to patterns of synaptic connectivity. Synaptic connectomes have proved difficult to interpret, in part due to their dense interconnectivity. If, as has been previously emphasized (*Bargmann and Marder, 2013*; *Brezina, 2010*; *Marder, 2012*), this interconnectivity reflects the multifunctionality of the underlying networks, and if the functional configuration of a network at any given time is determined by where and how neuromodulators are acting on its components, then patterns of neuromodulatory connectivity may provide a necessary complement to synaptic maps to render them interpretable. A key challenge will lie in identifying which neuromodulator systems play critical roles in establishing a network's output, but as the work here demonstrates, when these are known, the neuromodulatory connections can deliver substantial insight into how a neural network is organized.

# Materials and methods

## Key resources table

| Reagent type (species) or resource | Designation | Source or reference | Identifiers | Additional information |
|---|---|---|---|---|
| Genetic reagent (*D. melanogaster*) | ETHRB-Gal4 (ETHRB$^{MI00949}$-Gal4) | *Diao et al. (2016)* (doi: 10.1534/genetics.115.182121) | N/A | |
| Genetic reagent (*D. melanogaster*) | ETHRA-Gal4 (ETHRA$^{MI00949}$-Gal4) | *Diao et al. (2016)* (doi: 10.1534/genetics.115.182121) | N/A | |
| Genetic reagent (*D. melanogaster*) | ETHRA-p65AD (ETHRA$^{MI00949}$-p65AD) | *Diao et al. (2016)* (doi: 10.1534/genetics.115.182121) | N/A | |
| Genetic reagent (*D. melanogaster*) | ETHRB-p65AD | This paper | N/A | Split Gal4 hemidriver |
| Genetic reagent (*D. melanogaster*) | CCAP-R-Gal4 (CCAP-R$^{MI05804}$-GAL4) | *Diao et al. (2015)* (doi: 10.1016/j.celrep.2015.01.059) | N/A | |
| Genetic reagent (*D. melanogaster*) | CCAP-R-Gal4DBD (CCAP-R$^{MI05804}$-GAL4DBD) | This paper | N/A | Split Gal4 hemidriver |
| Genetic reagent (*D. melanogaster*) | CCAPR-p65AD (CCAP-R$^{MI05804}$-p65AD) | This paper | N/A | Split Gal4 hemidriver |
| genetic reagent (*D. melanogaster*) | CCAP-Gal4DBD | *Luan et al. (2006b)* (PMID: 17088209) | N/A | |
| Genetic reagent (*D. melanogaster*) | Burs-LexA::VP16AD | This paper | N/A | LexA driver |
| Genetic reagent (*D. melanogaster*) | RK-Gal4 (Rk$^{pan}$-Gal4) | *Diao and White (2012)* (doi: 10.1534/genetics.111.136291) | N/A | |
| Genetic reagent (*D. melanogaster*) | RK-Gal4DBD (Rk$^{TGEM}$-Gal4DBD) | This paper | N/A | Split Gal4 hemidriver |
| Genetic reagent (*D. melanogaster*) | RK-p65AD (Rk$^{TGEM}$-p65AD) | This paper | N/A | Split Gal4 hemidriver |
| Genetic reagent (*D. melanogaster*) | RK- LexA::QFAD (Rk$^{TGEM}$- LexA::QFAD) | This paper | N/A | Split Gal4 hemidriver |
| Genetic reagent (*D. melanogaster*) | VGlut-LexA::QFAD (VGlut$^{MI04979}$-LexA::QFAD) | *Diao et al. (2015)* (doi: 10.1016/j.celrep.2015.01.059) | N/A | |
| Genetic reagent (*D. melanogaster*) | VGlut-Gal4DBD (VGlut$^{MI04979}$-Gal4DBD) | *Diao et al. (2015)* (doi: 10.1016/j.celrep.2015.01.059) | N/A | |
| Genetic reagent (*D. melanogaster*) | UAS-GCaMP6S, insertions on Chromosomes II and III | Bloomington Drosophila Stock Center (BDSC) | 42746; 42749 | |
| Genetic reagent (*D. melanogaster*) | UAS-Kir2.1 insertions on Chromosomes II and III | Bloomington Drosophila Stock Center | 6596 | |
| Genetic reagent (*D. melanogaster*) | UAS-dTrpA1 | oth er | BDSC 26263 | Paul Garrity, Brandeis |
| Genetic reagent (*D. melanogaster*) | tubP-Gal80$^{ts}$-20 | Bloomington Drosophila Stock Center | 7019 | |
| Genetic reagent (*D. melanogaster*) | UAS-P2X2 | other | N/A | Orie Shafer, Univ. of Michigan |
| Genetic reagent (*D. melanogaster*) | MiMIC CCAP-R[MI05804] | Bloomington Drosophila Stock Center | BDSC 40788 | |
| Genetic reagent (*D. melanogaster*) | UAS-6XEGFP on II and III | Bloomington Drosophila Stock Center | 52261; 52262 | |
| Genetic reagent (*D. melanogaster*) | UAS-6XmCherry on III | Bloomington Drosophila Stock Center | 52268 | |
| Genetic reagent (*D. melanogaster*) | 24B (How)-Gal4 | Bloomington Drosophila Stock Center | 1767 | |
| Genetic reagent (*D. melanogaster*) | {nosCas9} attP2 line | *Ren et al. (2013)* (doi: 10.1073/pnas.1318481110) | | |

*Continued on next page*

*Continued*

| Reagent type (species) or resource | Designation | Source or reference | Identifiers | Additional information |
|---|---|---|---|---|
| Genetic reagent (*D. melanogaster*) | W[1118] | other | | White lab stock |
| Antibody | Rabbit polyclonal anti-pBurs | other | N/A | Aaron Hsueh/Willi Honegger, Used at 1:1000 |
| Antibody | Alexafluor555-conjugated guinea pig anti-mouse | Invitrogen | 1789887 | |
| Recombinant DNA reagent | U6b-sgRNA-short plasmid | *Ren et al. (2013)* (doi: 10.1073/pnas.1318481110) | | |
| Recombinant DNA reagent | pT-GEM(1) plasmid | *Diao et al. (2015)* (doi: 10.1016/j.celrep.2015.01.059) | | |
| Recombinant DNA reagent | pCAST-Burs[Gal4DBD] | *Luan et al. (2012)* (doi: 10.1523/JNEUROSCI.3707–11.2012) | | |
| Recombinant DNA reagent | pBS-KS-attB-SA-SD-0-T2A-P65AD vector | *Diao et al. (2015)* (doi: 10.1016/j.celrep.2015.01.059) | | |
| Recombinant DNA reagent | pBS-KS-ETHR[MI00949]-T2A-p65AD in 4B | This paper | | See *Supplementary file 1* |
| Sequence-based reagent | guide RNA oligos for Rk gene: ttcgTAAGTGAACCTTCAATGTCT; aaacAGACATTGAAGGTTCACTTA | Integrated DNA Technologies, Inc. | N/A | |
| Sequence-based reagent | PCR primers for Rk left homology arm: acccaccggaccggtgcatgCAAC CTCGACCCTTCAGTTCC; GACCTGGGGCGGCCGCG ctagacattgaaggttcacttac; | Integrated DNA Technologies, Inc. | N/A | |
| Sequence-based reagent | PCR primers for Rk right homology arm: cctgggggcgcgccggtacGGTA ATATTACATTAATTATTCTAAC; GAACCTCCCCACTAGTG gagaaagggattgcagcaac; | Integrated DNA Technologies, Inc. | N/A | |
| Sequence-based reagent | Drosophilized LexA::VP16AD construct | Epoch Life Science, Inc. | N/A | |
| Sequence-based reagent | PCR primers for T2A-P65AD forward: cgcgccagcaagatcgaggg ccgcggcagcctg PCR primers for T2A-P65AD reverse: atgggattcagatcttta cttgccgccgcccag | Integrated DNA Technologies, Inc. | N/A | |
| Peptide, recombinant protein | Ecdysis Triggering Hormone 1 (ETH1) | GenScript | P11731308 | |
| Commercial assay or kit | | | | |
| Chemical compound, drug | Alexa Fluor 594 Phalloidin | ThermoFisher, Scientific | A12381 | |
| Chemical compound, drug | ATP | Sigma | A9187 | |
| Software, algorithm | PhaseFinder | This paper | https://github.com/BenjaminHWhite/PhaseFinder | Detects pupal ecdysis Phases in Ca++ activity records |

## Reagents

ETH1 was synthetized by GenScript (Piscataway, NJ); all oligonucleotides were synthesized by Integrated DNA Technologies, Inc (Coralville, IA); and all gene synthesis were carried out by Epoch Life Science, Inc (Sugar Land, TX). All restriction enzymes were from New England Biolabs (Ipswich, MA).

## Fly lines

Vinegar flies of the species *Drosophila melanogaster* were used in this study. Flies were raised on cornmeal-molasses-yeast medium and housed at 25°C and 65% humidity. Both male and female progeny of the genotypes indicated in *Supplementary file 1* were used in this study and all experiments analyzed animals at the time of pupal ecdysis, approximately 12 hr after puparium formation. Fly stocks described in previous publications include: ETHRA-Gal4 (i.e. ETHRA$^{MI00949}$-Gal4), ETHRB-Gal4 (i.e. ETHRB$^{MI00949}$-Gal4), ETHRA-p65AD (i.e. ETHRA$^{MI00949}$-p65AD), all from *Diao et al. (2016)*; CCAP-R-Gal4 (CCAP-R$^{MI05804}$-Gal4), VGlut-Gal4DBD (i.e. VGlut$^{MI04979}$-Gal4DBD) and VGlut-LexA:: QFAD (VGlut$^{MI04979}$-LexA::QFAD), all from *Diao et al. (2015)*; Rk$^{pan}$-Gal4 (*Diao and White, 2012*); and CCAP-Gal4DBD (*Luan et al., 2006b*). The ETHRB-p65AD line was made by ΦC31-mediated cassette exchange into MiMIC insertion MI00949 in the ETHR gene using a strategy previously used to make the ETHRA-p65AD line (*Diao et al., 2016*). Briefly, we created an 'ETHRB$^{MI00949}$-p65AD in 4b' construct (*Supplementary file 2*) by combining two fragments: one was a PCR-generated fragment encoding T2A-p65AD, amplified from the pBS-KS-attB-SA-SD-0-T2A-P65AD vector (*Diao et al., 2015*) using the primers listed in the Key Resources Table, and the other was a synthesized gene fragment corresponding to the ETHR genomic region from the MI00949 insertion point to the 3' end of exon 4b. The latter fragment included an extension containing an Hsp70 polyadenylation signal and was flanked by Sal I restriction sites, which were used to subclone the fragment into the pBS-KS-attB1-2 vector (*Venken et al., 2011*). The T2A-p65AD fragment was inserted in frame into a unique *Bgl II* site just prior to the stop codon of Exon 4b in the synthesized fragment using the In-Fusion Cloning Kit from Takara Bio USA, Inc (Mountain View, CA). The resulting vector (pBS-KS-ETHR$^{MI00949}$-T2A-p65AD in 4B) was used for ΦC31-mediated transgenesis. All other lines created for use in this paper were generated using the Trojan exon technology by plasmid injection as described in *Diao et al. (2015)*. All injections were made by Rainbow Transgenic Flies, Inc (Camarillo, CA). CCAP-R-specific Split Gal4 lines (CCAP-R-Gal4DBD and CCAP-R-p65AD), were generated by inserting the indicated Trojan exons into the MI05804 MiMIC site in intron 4 of the CCAP-R gene. New Rk-specific lines were generated by first inserting the Trojan Gal4 Expression Module (T-GEM) into the Rk locus using Crispr/Cas technology as described previously (*Diao et al., 2015*). The guide RNA (sgRNA) used to target T-GEM insertion was specific for a PAM site in intron 13 of the Rk gene and was made by annealing the two oligos listed in the Key Resources Table. The sgRNA was then inserted into the U6b-sgRNA-short plasmid of *Ren et al. (2013)* after digestion with *Bbs I*. The T-GEM construct was flanked by left (HAL) and right (HAR) homology arms of approximately 1 kb in length amplified by PCR using the primers indicated in the Key Resources Table (where upper case indicates sequences homologous to Rk and lower case indicates sequence homologous to the pT-GEM(1) plasmid.) The PCR products were cloned into the linearized pT-GEM(1) plasmid using the In-Fusion Cloning Kit from Takara Bio USA, Inc (Mountain View, CA). The Rk$^{TGEM}$-Gal4 transgenic flies were made by microinjecting embryos of the {nosCas9} attP2 line (*Ren et al., 2013*) with the sgRNA and pT-GEM plasmid DNA. The microinjection was made by Rainbow Transgenic Flies, Inc (CA). The G0 adults were crossed with yw;Sp/Cyo;Dr/TM3,Sb flies and the progeny were screened by fluorescence for those with eye-specific expression of the selection marker, RFP. Rk-specific Split Gal4 hemidriver lines were subsequently generated from this Rk$^{TGEM}$-Gal4 line by substituting Gal4DBD and p65AD Trojan exons into the site of T-GEM insertion. The Burs-LexA::VP16AD line was generated from a drosophilized LexA::VP16AD DNA construct (*Supplementary file 2*) in pBlueScript synthesized by Epoch Life Science, Inc (Sugar Land, TX). Flanking *NotI* and *AscI* restriction sites in the construct were used to subclone this construct into the pCAST-Burs$^{Gal4DBD}$ plasmid (*Luan et al., 2012*) after excision of the Gal4DBD sequence. Burs-LexA::VP16AD flies were made by standard P-element transgenesis by injecting the resulting pCAST-Burs-LexA::VP16AD plasmid into the embryos of w$^{1118}$ flies and a line was established with the transgene inserted on Chromosome II. The UAS-dTrpA1 and UAS-P2X2 lines were the kind gifts of Paul Garrity and Orie Shafer, respectively. As indicated in the Key Resources Table, all other fly lines were obtained from the Bloomington Drosophila Stock Center at Indiana University including the MI05804 line, which was generated by the *Drosophila* Gene Disruption Project, http://flypush.imgen.bcm.tmc.edu/pscreen/mimic.html (*Nagarkar-Jaiswal et al., 2015*).

## Manipulation and monitoring of neuronal activity

All neuronal suppression experiments were conducted using two copies of UAS-Kir2.1, by combining insertions on Chromosomes II and III. To restrict suppression to the pupal stage, we used the temperature-sensitive Gal4 inhibitor, tsGal80, expressed under the control of the ubiquitous tubulin promoter (i.e. tub-Gal80ts) (*McGuire et al., 2003*). Animals were shifted to 31°C at the wandering L3 stage. Neuronal activation of ETHRA/CCAP neurons was accomplished using UAS-dTrpA1 using temperature shifts from 18°C to 29°C. Transient temperatures shifts were accomplished by heating to 29°C for 1 min before returning to 18°C.

## Behavioral analysis

The method used for videorecording of pupal ecdysis behaviors was described in *Diao et al. (2016)*. Briefly, cryptocephalic pupae were selected for videorecording just prior to pupal ecdysis, after the abdominal bubble had appeared and vigorous movement of the gut had commenced. Puparia were coated with a mixture of halocarbon oil and water (~2:1) to increase their transparency and placed ventral side down on a cover glass, which was attached to a glass slide with doublestick tape to form a small chamber. Behavior was recorded from the ventral side for 1–2 hr at 20X magnification using a Sony NEX VG20 camcorder mounted on an Olympus SZX12 stereomicroscope. Videorecords were imported into iMovie (Apple Inc., Cupertino, CA) software for behavioral analysis. Scoring of the pupal ecdysis phases largely followed the criteria described by *Kim et al. (2006)*: Phase I (pre-ecdysis) begins when the tip of the abdomen is lifted, creating an air pocket at the posterior end of the puparium, and continues with posterior-to-anterior 'rolling' contractions of the lateral body wall; Phase II (ecdysis) is characterized by persistent swinging, resulting from alternating lateral contractions of the abdomen; and Phase III (post-ecdysis) consists predominantly of anterior-to-posterior 'stretch-compressions' of the abdomen. The transition from Phase II to Phase III is not well-defined and consists of mixed abdominal movements that include intermingled swings and stretch-compressions. These various movements are often poorly resolved when viewed through the puparium, even when the latter are clarified by treatment with an oil/water mixture. To unambiguously define the end of Phase II for behavioral experiments, and to simplify the analysis, we therefore defined the last occurrence of any swinging (scored by playing the videos backwards and marking the time of the 'first' swinging bout) as the end of Phase II. For the muscle $Ca^{++}$ imaging experiments, where the transition from consolidated swinging to mixed abdominal movements could be resolved, we defined the end of consolidated swinging as the end of Phase II, and divided Phase III into a 'Transition Period' consisting of the mixed behavioral phase followed by a 'late Phase III' period of consolidated stretch-compressions. This division conformed well with the patterns of neural $Ca^{++}$ activity recorded in VNC-Rk and glutamatergic CCAP-R-expressing neurons.

## Immunohistochemistry

Nervous system whole mounts were excised from stage four pupae with an air bubble (*Bainbridge and Bownes, 1981*) and prepared for immunolabeling as described previously, using normal donkey or goat serum in the blocking solution (*Luan et al., 2006a*). Rabbit anti-pBurs (kind gift of Aaron Hsueh and Willi Honegger) was used at 1:1000 dilution. Guinea pig secondary antibodies were conjugated to Alexa Fluor 555 (Invitrogen, Carlsbad, CA). Expression patterns of Gal4 and Split Gal4 lines were visualized using either UAS-6XEGFP or UAS-6XmCherry. To visualize labeling of motor axons by CCAP-R-p65AD ∩ VGlut-Gal4DBD, wandering third instar larvae were briefly anesthetized under $CO_2$, immersed in 100% ETOH, and then pinned out and filleted from the dorsal side in PBS. The head and internal organs were removed before fixation and staining. Muscle was visualized using Alexa Fluor 594-conjugated phalloidin (Invitrogen, Carlsbad, CA). Preparations were incubated with phalloidin at 1:1000 dilution for 2 hr, followed by three 10 min washes. Confocal imaging was done using a Nikon C2 personal confocal microscope with a 20X/0.75 NA air objective. Unless otherwise noted, the images presented are maximum intensity projection images of a Z-stack collected through the entire preparation.

## GCaMP $Ca^{++}$ imaging

$Ca^{++}$ imaging of excised pupal nervous systems was carried out using conditions similar to those originally described by *Kim et al. (2006)*. Cryptocephalic pupae expressing two copies of UAS-

GCaMP6s (on Chromosomes II and III) under the control of Rk-Gal4, ETHRB-Gal4, or the Split Gal4 driver CCAP-R-P65AD $\bigcap$ vGlut-Gal4DBD were dissected under cold phosphate-buffered saline (137 mM NaCl, 2.7 mM KCl, 10 mM $Na_2HPO_4$ and 2 mM $KH_2PO_4$, pH 7.3) approximately 2 hr prior to pupal ecdysis as determined by the appearance of the abdominal gas bubble and the onset of gut movement. CNSs were excised and placed on poly-lysine (Sigma-Aldrich, St. Louis, MO) coated cover glass, dorsal side up, and then covered with Schneider's Insect Medium. GCaMP6s fluorescence in the VNC was imaged at a frequency of approximately 1 Hz for 90 min using a Nikon C2 confocal microscope with a 20X/0.75 NA air objective. Imaging was carried out using the largest pinhole (optical section thickness of 150 μm) and focusing on a plane approximately 20–35 μm below the dorsal surface, to maximize the number of neurons in the field. Preparations were imaged for 5 min to measure baseline $Ca^{++}$ activity prior to addition of 600 nM synthetic ETH1.

Muscle $Ca^{++}$ activity was imaged in pupal animals that expressed two copies of UAS-GCaMP6s under the control of the muscle driver 24B-Gal4. These animals were selected prior to pupal ecdysis as described above, and their puparia were rinsed with 50% bleach for 3 min to permit optimal clarity. Puparia were then mounted as described for behavioral videorecording in a halocarbon oil/water mixture, and GCaMP6s florescence was imaged from the ventral side at a frequency of approximately 1 Hz for 90 min using a Nikon C2 confocal microscope with a 4X/0.13 NA air objective. By using the largest pinhole and a single image plane, $Ca^{++}$ activity of most bodywall muscles could be visualized.

## Fluorescence image processing

The Nikon C2 imaging files collected during GCaMP imaging were saved as ND2 files for quantification. All ND2 format images were background-subtracted and mean intensities over the regions of interest (ROI) measured using ImageJ software (*Schneider et al., 2012*). ROI are as indicated in the figures and $Ca^{++}$ traces were then normalized to the average signal (F) measured during the initial 5-min period prior to addition of ETH and are presented as ΔF/F in the figures. Muscle $Ca^{++}$ traces were normalized to the average signal measured over the first 250 frames. For the experiment shown in *Figure 5—figure supplement 1*, the frequencies of $Ca^{2+}$ oscillations for single ROIs were determined using the PhaseFinder program (described below) and average frequencies were calculated for each Phase, the duration of which was calculated by PhaseFinder using the global $Ca^{++}$ signal for the Rk-VNC neurons. Heatmaps showing the average Phase frequencies for each individual ROI (and the whole population) were created with MatLab using the built-in heatmap function. Movies prepared for presentation were exported as. avi files and then converted into iMovie format for editing and display.

## Behavioral Phase detection using PhaseFinder

Custom MatLab code (available at GitHub: https://github.com/BenjaminHWhite/PhaseFinder) was written to objectively identify the onset of each phase of $Ca^{++}$ activity. This was achieved as shown schematically in *Figure 4—figure supplement 1*. Using ImageJ, the mean $Ca^{++}$ activity within an ROI drawn over the whole VNC was calculated for each image in a timeseries to generate a $Ca^{++}$ trace. The PhaseFinder code operated on such a trace by first identifying peaks, and then a sliding window was used to compare the frequency or average amplitude of the peaks in one time-window to those in the next. The onset of Phase 1 was defined as the time of the first peak of activity following the addition of ETH. To find the onset of Phase 2 (and offset of Phase 1), the difference in average peak amplitude was calculated for consecutive windows and the first window for which this difference was more than one standard deviation above the average peak amplitude across all windows was defined as the beginning of Phase 2. Phase 3 was the most difficult to define because of the complexity of its $Ca^{++}$ activity patterns, but it was generally distinguished from the two preceding phases by a change in peak frequency. As described in the text, we divided this phase into a 'Transition Period,' of mixed frequencies, and a period of more uniform frequency ('late Phase 3'). The onset of Phase 3 (which also marked the offset of Phase 2) was defined by the decrement in frequency that occurred at the time of phase transition. This was most easily identified by analyzing the time series data in reverse, using time windows starting at the end of the $Ca^{++}$ trace and looking for the first window for which the difference in frequency was positive (indicating a decrement in frequency moving forward in time). The uniformity of activity in late Phase 3 allowed its onset (and the

offset of the Transition Period of Phase 3) to be defined simply as the first window for which the difference in peak frequency exceeded the average peak frequency for all windows by more than one standard deviation. Phase durations were calculated by subtracting the onset of each phase from the onset of the next phase. Further details about each parameter used in the PhaseFinder program can be found in the documentation available at Github (see below). Importantly, the parameter settings used to analyze all neuronal datasets (i.e. for both Rk- and CCAP-R-expressing neurons) were the same. Muscle $Ca^{++}$ traces required somewhat different parameters because of the significantly larger signal amplitudes in integrated mean fluorescence.

## Sample sizes and statistics

For all experiments, the number of biological replicates (i.e. the number of animals or CNS preparations of a given genotype) analyzed was at least five, with the actual numbers for each experiment given in the figures or figure legends. Estimation of required sample sizes were made using the procedure of *Campbell et al. (1995)* for binary categorical variables, since most experiments involved determining whether manipulation of a specific neuronal population (e.g. one expressing the receptors for Bursicon or CCAP) resulted in pupal ecdysis deficits or not. Similar manipulations previously applied to neurons expressing Bursicon (*Peabody et al., 2008*) or CCAP (*Park et al., 2003a*) report effect sizes on pupal ecdysis ranging from approximately 0.45 to 0.9, indicating the use of samples sizes of 4 to 10. The correlation analyses of $Ca^{++}$ signals measured on the left- and right-hand sides of the midline for VNC-Rk neurons and muscles were performed in MatLab using the built-in Corrcoeff function. For the statistical analysis of $Ca^{++}$ oscillation frequencies in *Figure 6E*, GraphPad Prism was used to conduct a one-way ANOVA with multiple comparisons. Brown-Forsythe and Bartlett's tests determined that the variance between groups was not significant and a Sidak's multiple comparison's test showed no significant differences between Rk and CCAPR-Vglut frequencies at any of the phases.

## Acknowledgements

This work was supported by the Intramural Research Program of the National Institute of Mental Health (ZIAMH002800, BHW) and by a Postdoctoral Research Associate (PRAT) Fellowship from the National Institute of General Medical Sciences (FI2 GM117582, AE). We further thank the Bloomington Drosophila Stock Center (NIH P40OD018537) for many of the fly stocks used in this study, and Quentin Gaudry for insightful comments on the original manuscript. Finally, we thank Eve Marder for encouraging us to frame the work in a broader historical context.

## Additional information

### Funding

| Funder | Grant reference number | Author |
| --- | --- | --- |
| National Institute of Mental Health | MH002800-15 | Benjamin H White |
| National Institute of General Medical Sciences | FI2 GM117582 | Amicia D Elliott |

The funders had no role in study design, data collection and interpretation, or the decision to submit the work for publication.

### Author contributions

Feici Diao, Conceptualization, Investigation, Visualization, Methodology, Writing—original draft, Writing—review and editing; Amicia D Elliott, Conceptualization, Software, Formal analysis, Supervision, Funding acquisition, Investigation, Visualization, Methodology, Writing—review and editing; Fengqiu Diao, Resources, Methodology; Sarav Shah, Software, Formal analysis; Benjamin H White, Conceptualization, Resources, Supervision, Funding acquisition, Visualization, Methodology, Writing—original draft, Project administration, Writing—review and editing

Author ORCIDs
Benjamin H White (iD) http://orcid.org/0000-0003-0612-8075

Decision letter and Author response
Decision letter https://doi.org/10.7554/eLife.29797.027
Author response https://doi.org/10.7554/eLife.29797.028

## Additional files

### Supplementary files

• Supplementary file 1. Fly genotypes used listed by figure.
DOI: https://doi.org/10.7554/eLife.29797.023

• Supplementary file 2. Sequences of DNA constructs used to make ETHRB-p65AD and Burs-LexA::VP16AD lines.
DOI: https://doi.org/10.7554/eLife.29797.024

• Transparent reporting form
DOI: https://doi.org/10.7554/eLife.29797.025

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
