## [Decision Letter]

Thank you for submitting your article "Neuromodulatory connectivity defines the structure of a behavioral neural network" for consideration by *eLife*. Your article has been reviewed by three peer reviewers, one of whom, Ronald L Calabrese (Reviewer #3), is a member of our Board of Reviewing Editors, and the evaluation has been overseen by Eve Marder as the Senior Editor. The following individual involved in review of your submission has agreed to reveal their identity: Michael E Adams (Reviewer #2).

The reviewers have discussed the reviews with one another and the Reviewing Editor has drafted this decision to help you prepare a revised submission.

Summary:

This is an elegant and informative study that advances our knowledge considerably regarding neuromodulatory regulation of the pupal ecdysis behavioral sequence in *Drosophila melanogaster*. The authors have capitalized on their earlier development of Trojan exon targeting of peptide receptors to identify neuronal targets of ETH, bursicon, and CCAP and ascertain their roles in each of the sequential behaviors. Using a combination of behavioral monitoring, calcium imaging, and selective activation of neuronal groups through expression of temperature sensitive TrpA1 and ATP/P2X2, they have identified layers (modular input, CPG, motor output) of circuit organization and how they interact to produce an innate sequence of behaviors. These findings have general relevance to our understanding of how multiple centrally patterned behaviors can be encoded by modulating the properties/connections of neural circuit(s). The work is nicely illustrated with exemplary data. The writing is clear and the Introduction in particular very scholarly.

Essential revisions:

1) The manuscript as written does not provide a clear description of the genotypes used in the experiments. This makes it difficult at times to gauge the quality of experimental controls. It is also a significant barrier to future replication of the experiments. Full genotypes, preferably using standard *Drosophila* genetic notation, should be included in the manuscript in a way that makes it easy for the reader to determine the precise genotypes used for each experiment. This is particularly pressing for the first P2X2 experiment shown in Figure 3, as it is not completely clear what the genotype of the control line is here (see point two below).

2) The P2X2 functional connectivity experiments described appear to use inconsistent negative controls. In the first (Figure 3,) ATP is applied to preparations that lack P2X2 expression. In the second and third (Figure 5 and Figure 6), the negative control is the same time-course imaging and genotype in the absence of ATP application. In order to fully control this experiment, it is important to apply ATP to a preparation that contains the P2X2 responder element without its genetic driver. This controls for both the potential effects of ATP application and for the possibility of leaky P2X2 expression that may occur independently of the genetic driver used in the experimental animals. It is not clear if this is the control used in Figure 3, but it seems clear that this was not what was done for Figure 5 and 6. The P2X2 controls are straightforward and critical. Now that this method has been used more extensively, it has become clear that leaky P2X2 can result in the direct ATP mediated excitation of the presumptive target. The authors need only repeat the ATP application experiments on genotypes that lack the driver for P2X2 Expression for last two P2X2 experiments. This would require no new reagents and would take about a month and a half, most of which would be waiting on the flies to eclose.

3) The level of quantification and analysis of the Ca-imaging time series is a bit disappointing. The authors seem to be focused on showing the match between neuronal activity, muscle activity, and behavior but miss the opportunity to provide in-depth analysis of the rhythms/patterns observed. Particularly for the data of Figure 5 and Figure 5—figure supplement 1 there is a missed opportunity to present neuronal phase data using modern phase analyses. The data of Figure 5—figure supplement 1 are used to justify the argument that the GPGs are multifunctional but the analyses are not deep. Several of the studies cited do provide such phase analyses, e.g. Briggman and Kristan, 2006; 2008, not just correlations, average behavior-phase durations and average frequencies. We ask that the authors consider expanding these analyses.

---

## [Author Response]

Essential revisions:1) The manuscript as written does not provide a clear description of the genotypes used in the experiments. This makes it difficult at times to gauge the quality of experimental controls. It is also a significant barrier to future replication of the experiments. Full genotypes, preferably using standard Drosophila genetic notation, should be included in the manuscript in a way that makes it easy for the reader to determine the precise genotypes used for each experiment. This is particularly pressing for the first P2X2 experiment shown in Figure 3, as it is not completely clear what the genotype of the control line is here (see point two below).

We agree entirely and apologize for this omission. We had, in fact, intended to include a comprehensive list of the genotypes of all animals used in the experiments presented, but inadvertently neglected to do so. The revised submission now includes this list as Supplementary file 1.

2) The P2X2 functional connectivity experiments described appear to use inconsistent negative controls. In the first (Figure 3,) ATP is applied to preparations that lack P2X2 expression. In the second and third (Figure 5 and Figure 6), the negative control is the same time-course imaging and genotype in the absence of ATP application. In order to fully control this experiment, it is important to apply ATP to a preparation that contains the P2X2 responder element without its genetic driver. This controls for both the potential effects of ATP application and for the possibility of leaky P2X2 expression that may occur independently of the genetic driver used in the experimental animals. It is not clear if this is the control used in Figure 3, but it seems clear that this was not what was done for Figure 5 and 6. The P2X2 controls are straightforward and critical. Now that this method has been used more extensively, it has become clear that leaky P2X2 can result in the direct ATP mediated excitation of the presumptive target. The authors need only repeat the ATP application experiments on genotypes that lack the driver for P2X2 Expression for last two P2X2 experiments. This would require no new reagents and would take about a month and a half, most of which would be waiting on the flies to eclose.

The point here is well taken and, again, we apologize for the omission of the genotype data. As indicated on the list of genotypes in Supplementary file 1, the controls requested were, in fact, conducted for the experiments shown in Figure 3 and Figure 6, thus excluding possible ATP responses by both ectopically expressed P2X2 and endogenously expressed ATP receptors. The latter possibility was also controlled for in the experiment shown in Figure 5, though in this case we did not control for possible leaky expression of P2X2. We were at the time of the experiment unaware of the report that ectopically expressed P2X2 could give rise to spurious signals. Because of the difficulty of genetically recombining the necessary transgenes, we were also unable to repeat this experiment within the two-month period of revision. It thus remains possible that ectopic P2X2 expression induces the observed Ca^++^ response in the VGlut-expressing neurons. We note, however, that if this is true, the responding population cannot include the neurons that also express CCAP-R based on the negative results obtained with the control preparations shown Figure 6. This seems less likely to us than the alternative possibility, which is that the same population of VGlut- and CCAP-R-expressing neurons is responding to activation of upstream Rk neurons in the two experiments. Because of the consistency of the results from the experiments in Figure 5 and Figure 6, we are loathe to omit Figure 5, but we will do so if the reviewers think it necessary. We note, however, that the control performed in Figure 5 has been an accepted standard in the literature (and, in fact, was used in a paper published earlier this year in *eLife*: doi: 10.7554/*eLife*.23206), and that the reviewers’ published comments might serve as sufficient warning to readers that our results are open to this caveat. In response to the reviewers’ comments we have also amended the legends to Figure 3, Figure 5, and 6 to clarify the nature of the control experiments.

3) The level of quantification and analysis of the Ca-imaging time series is a bit disappointing. The authors seem to be focused on showing the match between neuronal activity, muscle activity, and behavior but miss the opportunity to provide in-depth analysis of the rhythms/patterns observed. Particularly for the data of Figure 5 and Figure 5—figure supplement 1 there is a missed opportunity to present neuronal phase data using modern phase analyses. The data of Figure 5—figure supplement 1 are used to justify the argument that the GPGs are multifunctional but the analyses are not deep. Several of the studies cited do provide such phase analyses, e.g. Briggman and Kristan, 2006; 2008, not just correlations, average behavior-phase durations and average frequencies. We ask that the authors consider expanding these analyses.

As noted above, we are very sympathetic with the reviewers on this point. We’d love to be able to carry out the type of analysis that Briggman and Kristan conducted on the leech swimming and crawling CPGs! In fact, our hope is that our work will pave the way for a comprehensive analysis of not only the CPGs, but of all the relevant components of the pupal ecdysis network lying between the sites of hormonal input and the motor neurons. Before that is possible, however, we will need to be able to meet two conditions that facilitated Briggman and Kristan’s elegant analysis. First, we need to be able to confidently assign optically observed Ca^++^ changes to individually identifiable neurons, and second, we must be able to independently monitor fictive motor program output using the activity of identified neurons as proxies. Using our current imaging paradigm, the spatial resolution of our Ca^++^ activity records is relatively limited, particularly in the z-dimension, which makes obtaining bona fide single cell traces difficult, if not impossible. In addition, we lack a gold standard for following the motor output. We have recently completed construction of a light-sheet microscope that will permit two-channel imaging. Our goal is to collect brain-wide measurements of single-cell responses to ETH from neurons in different layers of the pupal ecdysis network while simultaneously recording motor neuron activity as a proxy for behavior from the same preparation. We think that this type of data will lend itself well to the type of deep analysis that the reviewers request and hope that they will view our failure to provide it not so much as a “missed opportunity,” but as an opportunity prudently deferred.

Given the limitations of our current imaging capabilities, we felt that the analysis shown in Figure 5—figure supplement 1 of our original manuscript was about as deep as our data warranted. Our intention was to provide suggestive, rather than conclusive, evidence for the multifunctionality of the putative CPG neurons in the VNC-Rk pattern and we are still inclined to think that this analysis is worth including. However, the reviewers’ disappointment with its limited scope indicates that our original treatment needs to be improved, if only to limit the reader’s expectations. Accordingly, we have tried in the revised manuscript to: 1) clearly spell out the limitations of our approach, explicitly stating in the Materials and methods our limited imaging resolution in the z-dimension (see “GCaMP Ca^++^ imaging”); 2) indicate that the analysis is “preliminary”; and 3) more carefully qualify our conclusions. The text of both the Results and the legend of the revised figure (i.e. “Figure 5—figure supplement 1”) now make it clear that the small ROIs selected for analysis cannot necessarily be interpreted as “neurons.” In addition, after considering the reviewers’ comments, we thought it might be useful to summarize in some way our analysis of the 95 selected small ROIs rather than simply present selected Ca^++^ traces. Although no single parameter defines the phases of Ca^++^ activity, oscillation frequency is an important determinant. For each ROI, we therefore calculated the average oscillation frequency for each Phase (defined from the global Ca^++^ signal over the entire VNC-Rk population) using the PhaseFinder program. The activity of each ROI can thus be represented as a set of average Phase frequencies that can be compared across ROIs and against the global average. This data—represented as a heatmap—is now included as panel C in “Figure 5—figure supplement 1. Represented in this way, one can see that the ROIs exhibit a heterogeneous set of activity patterns, but that roughly 10% of them have frequency patterns across Phases that resemble that of the whole population—again consistent with the conclusion that some neurons within the VNC-Rk pattern may be multifunctional.

We realize that this additional analysis is by no means definitive proof that VNC-Rk neurons constitute a multifunctional central pattern generator and that further analysis will be required. To lay the groundwork for such an analysis, however, the first step has to be the plausible identification of the CPG neurons. Our goal was to provide such a demonstration, not only for the putative CPG neurons, but also for the neurons of the input and motor output layers. One of the strengths of the manuscript as we see it is that it provides a map of where to look for the different functional components of the pupal ecdysis network by revealing the molecular markers that distinguish them. By providing such a map, we not only confirm the insight derived from the pioneering work of Mike Adams’ lab—namely that ETH induces the generation of a bona fide fictive ecdysis sequence in the excised pupal nervous system—but open the door to a comprehensive analysis of how that sequence is generated from start to finish.